# Molecular Mechanism by Which Cobra Venom Cardiotoxins Interact with the Outer Mitochondrial Membrane

**DOI:** 10.3390/toxins12070425

**Published:** 2020-06-27

**Authors:** Feng Li, Indira H. Shrivastava, Paul Hanlon, Ruben K. Dagda, Edward S. Gasanoff

**Affiliations:** 1STEM Program, Science Department, Chaoyang KaiWen Academy, Yard 46, 3rd Baoquan Street, Chaoyang District, Beijing 100018, China; lifeng85315@126.com (F.L.); Paul.Hanlon@cy.kaiwenacademy.cn (P.H.); edward.gasanoff@cy.kaiwenacademy.cn (E.S.G.); 2Department of Computational and System Biology, University of Pittsburgh, Pittsburgh, PA 15260, USA; ihs2@pitt.edu; 3Department of Environmental and Occupational Health, University of Pittsburgh, Pittsburgh, PA 15260, USA; 4Reno School of Medicine, Department of Pharmacology, University of Nevada, Reno, NV 89557, USA

**Keywords:** cobra cardiotoxins, non-bilayer immobilized phospholipids, outer mitochondrial membrane, cardiolipin, phosphatidylcholine, ATP-synthase

## Abstract

Cardiotoxin CTII from *Naja oxiana* cobra venom translocates to the intermembrane space (IMS) of mitochondria to disrupt the structure and function of the inner mitochondrial membrane. At low concentrations, CTII facilitates ATP-synthase activity, presumably via the formation of non-bilayer, immobilized phospholipids that are critical in modulating ATP-synthase activity. In this study, we investigated the effects of another cardiotoxin CTI from *Naja oxiana* cobra venom on the structure of mitochondrial membranes and on mitochondrial-derived ATP synthesis. By employing robust biophysical methods including ^31^P-NMR and ^1^H-NMR spectroscopy, we analyzed the effects of CTI and CTII on phospholipid packing and dynamics in model phosphatidylcholine (PC) membranes enriched with 2.5 and 5.0 mol% of cardiolipin (CL), a phospholipid composition that mimics that in the outer mitochondrial membrane (OMM). These experiments revealed that CTII converted a higher percentage of bilayer phospholipids to a non-bilayer and immobilized state and both cardiotoxins utilized CL and PC molecules to form non-bilayer structures. Furthermore, in order to gain further understanding on how cardiotoxins bind to mitochondrial membranes, we employed molecular dynamics (MD) and molecular docking simulations to investigate the molecular mechanisms by which CTII and CTI interactively bind with an *in silico* phospholipid membrane that models the composition similar to the OMM. In brief, MD studies suggest that CTII utilized the N-terminal region to embed the phospholipid bilayer more avidly in a horizontal orientation with respect to the lipid bilayer and thereby penetrate at a faster rate compared with CTI. Molecular dynamics along with the Autodock studies identified critical amino acid residues on the molecular surfaces of CTII and CTI that facilitated the long-range and short-range interactions of cardiotoxins with CL and PC. Based on our compiled data and our published findings, we provide a conceptual model that explains a molecular mechanism by which snake venom cardiotoxins, including CTI and CTII, interact with mitochondrial membranes to alter the mitochondrial membrane structure to either upregulate ATP-synthase activity or disrupt mitochondrial function.

## 1. Introduction

We have shown that *Naja oxiana* cobra venom cardiotoxins CTII and CTI, which are more than 90% homologous to each other, specifically target cardiolipin (CL) in mitochondrial membranes and in model membranes composed of various phospholipids [1]. We have also shown that CTII can translocate into the intermembrane space (IMS) and facilitate the transition of bilayer phospholipids to a non-bilayer, immobilized state in the inner mitochondrial membrane (IMM) [1], a phospholipid membrane composed of at least 20% of CL in molar units [2]. In our previously published study, we observed that CTII possesses a remarkable ability to increase mitochondrial ATP-synthase activity in isolated bovine mitochondria, presumably through the induction of polymorphic transitions modulated by non-bilayer immobilized phospholipids which facilitate intermembrane contacts in mitochondrial cristae to enhance ATP-synthase activity [1,3,4,5]. Interestingly, the bio-physicochemical properties of CTII were observed to be similar to the cobra venom cardiotoxin VII4 from *Naja mossambica* cobra venom which can translocate and bind to mitochondria and alter the bioenergetic status of neuroblastoma cells and primary cortical neurons, presumably by binding to CL, but not PC, in the IMM leading to the formation of transient non-bilayer, immobilized toroidal structures [6]. So far, our findings suggest that snake venom cardiotoxins interact with mitochondria to alter the mitochondrial structure to either upregulate or downregulate oxidative phosphorylation. Our studies raise the possibility that snake venom cardiotoxins can be employed to modulate cellular bioenergetics in a variety of cell types. Therefore, treating tissues with a low concentration of cardiotoxins, known to bind CL in mitochondria, may represent a pharmacological tool to delay neurodegeneration by increasing the ATP production in neurons [6]. However, the molecular mechanism by which cardiotoxins interact with the OMM and translocate to the IMM to modulate mitochondrial-ATP synthesis is not clear [3,4,5,6].

In this article, we present robust biophysical and molecular dynamics data that shed insight on how a second cardiotoxin, namely CTI from *Naja oxiana* cobra venom, which also modulates mitochondrial ATP-synthase activity, can bind to the OMM to alter mitochondrial structure/function. By running parallel experiments, a comparative analysis of CTI and CTII suggests that both cardiotoxins can increase ATP synthesis via complex V in isolated mitochondria, with CTI showing a decreased ability to enhance ATP synthesis and form non-bilayer structures composed of immobilized phospholipids compared with CTII. By employing ^31^P-NMR and ^1^H-NMR spectroscopy and molecular dynamics/docking studies, we analyzed the ability of CTI and CTII to bind, penetrate and alter the organization of phospholipid bilayers. In brief, our compiled biophysical and in silico data suggest that both cardiotoxins interact with CL and PC in model membranes containing 2.5 and 5.0 mol% CL and can induce the generation of immobilized, non-bilayer packed structures presumably by binding to at least one CL molecule. Furthermore, the molecular docking experiments analysis study suggests that the molecular surface of CTII is predicted to harbor four CL and three PC binding sites, whereas CTI is predicted to contain three CL lipid binding sites and one lipid biding site for PC. Finally, we provide a conceptual model that provides insight on a unifying molecular mechanism by which venom cardiotoxins target and penetrate the OMM to translocate to the IMM to alter mitochondrial-ATP production.

## 2. Results

### 2.1. Cardiotoxins CTI and CTII Induce the Formation of Non-Bilayer Immobilized Phospholipids in Mitochondrial Membranes and Increase ATP-Synthase Activity

We have previously shown that phospholipids in mitochondria can transition between three different states under physiological conditions [1,3,4,5]: (1) a lamellar phase containing an anisotropic orientation of phosphodiester bonds of phospholipids packed in a bilayer organization, which is present in most biological membranes [1,3,4,5,7,8], (2) a non-bilayer lipid phase containing an isotropic orientation of phosphodiester bonds of phospholipids with restricted mobility also termed non-bilayer immobilized phospholipids [1,3,4,5], and (3) a non-bilayer packed lipid phase with rapid isotropic mobility of phosphodiester bonds of phospholipids (1 × 10^−2^–10^−4^ s) [3,4,5,7,8]. Consistent with these observations, here we show that a proton decoupled ^31^P-NMR spectrum of isolated bovine mitochondria recorded at 18 °C had a line shape that is consistent with the existence of phospholipid membranes in three biophysical states as previously published (Figure 1a) [1,3,4,5].

An asymmetrically shaped peak located at the high-field side of the spectrum at a resonance frequency of 11.9 ppm is presumably generated by phosphodiester bonds of phospholipids found in a lamellar phase [1,3,4,5,7,8]. A relatively narrow peak observed with a resonance frequency at 0.0 ppm (Figure 1a, under the arrow shown in letter (**i**) in the line shape) is a biophysical characteristic of non-bilayer structures that contain packed phospholipids of rapid isotropic mobility [1,3,4,5,7,8]. In Figure 1, a vertical line is drawn through a broad peak **B** at the resonance frequency 6.5 ppm. This broad peak signal **B** is derived from the non-bilayer immobilized phospholipids which presumably interact directly with the F_0_ subunit of ATP synthase and play an important role in facilitating ATP synthase activity [1,3,4,5]. Overall, per our previously published studies, we believe that this ^31^P-NMR spectrum line shape represents mitochondria containing intact phospholipid membranes (OMM and IMM) under physiological conditions [1,3,4,5]. 

Secondly, the addition of purified CTI and CTII to isolated bovine crude mitochondrial fractions enhanced the height of resonance signals emanating at a frequency of 0.0 and 6.5 ppm which represent phospholipids with rapid isotropic mobility and non-bilayer immobilized phospholipids, respectively (Figure 1b,c). The increase in the amplitude of the ^31^P-NMR signals induced by CTI and CTII was associated with a significant increase in ATP-synthase activity. Based on our previous studies, it is known that non-bilayer immobilized phospholipids, but not phospholipids with rapid isotropic mobility, are responsible for an increase in ATP synthase activity [1,3,4,5,6]. In addition, non-bilayer immobilized phospholipids, as opposed to phospholipids with rapid isotropic mobility, do not exchange with phospholipids found in a lamellar phase [1,3,4,5]. Therefore, by applying a DANTE train of saturation pulses at the high-field peak of the lamellar spectrum, we were able to determine the amount of immobilized phospholipids in isolated mitochondria treated with CTI or CTII (Figure 1, see arrow with letter S). The ^31^P-NMR signal located at a frequency of 6.5 ppm that remained following the application of a DANTE train in mitochondria is represented by a hatched line for all experimental conditions (Figure 1a–c). It is worth noting that the ^31^P-NMR signal represented by the hatched line is derived from non-bilayer, immobilized phospholipids and not from phospholipids with rapid isotropic mobility or phospholipids of a lamellar phase. We estimated the percentage of non-bilayer immobilized phospholipids in isolated mitochondria that remained following a DANTE pulse by calculating the area under the hatched line. CTII induced a higher percentage of non-bilayer immobilized phospholipids compared with CTI-treated mitochondria. In mitochondria, the population of non-bilayer immobilized phospholipids has been documented to arise from interaction with mitochondrial membrane proteins, presumably by interacting with the F_0_ subunit of ATP synthase [3,4,5,6]. However, in mitochondria treated with CTII or CTI, non-bilayer immobilized phospholipids can arise not only from the interaction of phospholipids with the F_0_ subunit of ATP synthase, but also by interacting with CTII or CTI. Thus, it appears that cardiotoxins facilitate ATP synthase activity by increasing the population of non-bilayer immobilized phospholipids. Given that CTII induces a higher amount of non-bilayer immobilized phospholipids relative to CTI per our biophysical data, it is conceivable that CTII enhances ATP-synthase activity to a greater extent than CTI by stimulating the formation of non-bilayer, immobilized structures.

### 2.2. Cardiotoxins CTII and CTI Induce Formation Non-Bilayer Structures in PC Liposomes Enriched with 2.5% mol and 5.0% mol CL

In order to model the interaction of cardiotoxins with the OMM, we examined the biophysical effects of CTII and CTI on the multilamellar liposomes of PC enriched with either 2.5 or 5.0 mol% CL. In addition, a sample of multilamellar liposomes exclusively consisting of PC was used as a control. To this end, we employed three samples of liposomes consisting of pure PC, containing PC and 2.5 mol% CL or containing PC + 5.0 mol% CL. In brief, the ^31^P-NMR spectra of all three samples of liposomes in the absence of cardiotoxins showed spectral lines that are characteristic of phospholipid membranes containing a lamellar packing of phospholipids (spectra are not shown). Moreover, the ^31^P-NMR spectra from the pure PC liposomes treated with CTII or CTI had an asymmetrical shaped peak located at the high-field side of the spectrum, which is consistent of phospholipid membranes containing a lamellar (bilayer) packing of phospholipids (Figure 2), suggesting that CTII and CTI had no biophysical effects on the pure PC liposomes.

However, in PC liposomes with either 2.5 or 5.0 mol% CL and treated with either CTII or CTI, we observed that the ^31^P-NMR spectra contained signals from two types of non-bilayer phospholipids, one with rapid isotropic mobility of phospholipids at 0 ppm and another with immobilized non-bilayer packed phospholipids through which a thin vertical line is drawn at around 4.9 ppm (Figure 2, signal **B**). By quantifying the signal intensities of both peaks, we observed that CTI induced a higher level of phospholipids with rapid mobility relative to CTII, whereas CTII induced a higher level of immobilized phospholipids compared with CTI (Figure 2). In a similar manner as shown in Figure 1, we estimated the percentage of non-bilayer immobilized phospholipids by calculating the area under the spectrum that remained after a DANTE train of saturation was applied at the high-field peak **S** of the lamellar spectrum (saturated spectrum not shown). Hence, by using this method, we calculated the number of moles of immobilized phospholipids per mole of CTII or CTI (Figure 2). A single molecule of CTII can immobilize eight phospholipids in multilamellar membranes containing PC and 5.0 mol% CL, and six phospholipids in multilamellar liposomes containing PC and 2.5 mol% CL liposomes. In addition, we calculated that CTI molecules can immobilize five phospholipids in multilamellar liposomes consisting of PC and 5.0 mol% CL and four phospholipids in multilamellar liposomes consisting of PC and 2.5 mol% CL.

### 2.3. Cardiotoxins CTII and CTI Involve PC Molecules in the Formation of Non-Bilayer Immobilized Phospholipid Structures

Based on our previous studies and studies of other researchers, ^1^H-NMR spectroscopy of unilamellar liposomes bathed in potassium ferricyanide K_3_[Fe(CN)_6_] is a biophysical technique that has been extensively used to study the ability of cardiotoxins, and other snake venom proteins, for permeabilizing and altering the structure of phospholipid bilayers [1,3,4,5,6,7,8,9,10,11,12,13,14]. In brief, ferricyanide ion, Fe(CN)_6_^3^^−^, interacts with ^1^H atoms of the choline groups N^+^(CH_3_)_3_ of PC on the outer leaflet of liposomes and shifts the ^1^H-NMR signal derived from the outer leaflet N^+^(CH_3_)_3_ groups towards a higher field, whereas the ^1^H-NMR signal derived from the N^+^(CH_3_)_3_ groups located in the inner leaflet remains unchanged given that the inner leaflet is not accessible for Fe(CN)_6_^3^^−^. Splitting ^1^H-NMR signals from the outer and inner leaflets induced by ferricyanide ion (Figure 3A) facilitates the quantitative analysis of the structural integrity and permeability of liposomes treated by cardiotoxins or other experimental conditions. Therefore, changes in liposome membrane structure, which allows Fe(CN)_6_^3^^−^ to permeate liposomal membrane and gain access to N^+^(CH_3_)_3_ groups in the inner leaflet, make the ^1^H-NMR signals from the outer and inner leaflets indistinguishable as they resonance now at the same frequency. We have consistently observed that the formation of non-bilayer immobilized phosphatidylcholines in the membrane of liposomes changes the ^1^H-NMR spectrum line shape of liposomes in Fe(CN)_6_^3^^−^ solution by giving rise to an additional ^1^H-NMR signal located on a high-field side from a signal derived from the N^+^(CH_3_)_3_ groups in the outer leaflet [13,14]. This phenomenon occurs due to the formation of non-bilayer structures containing immobilized phosphatidylcholines, which permits the closer interaction between the N^+^(CH_3_)_3_ groups and Fe(CN)_6_^3^^−^ in non-bilayer phospholipids containing PC, which further shifts the resonance frequency of the N^+^(CH_3_)_3_ groups to a higher field in the line spectrum.

Hence, by employing this well-validated biophysical technique, we investigated the extent to which CTII and CTI permeabilize PC liposomes containing either 2.5 or 5.0 mol% CL as well as analyzed the extent to which PC molecules are involved in the cardiotoxin-mediated formation of non-bilayer immobilized phospholipids observed by ^31^P-NMR spectroscopy. In brief, we observed that adding CTII to liposomes composed exclusively of PC did not affect the line shape of the ^1^H-NMR spectrum (Figure 3A). Like CTII, treating PC liposomes with CTI did not alter the line shape of the ^1^H-NMR spectrum (data not shown).

However, both CTI and CTII induced the formation of a high-field peak (indicated by the arrow in Figure 3B,C) on the right side from the outer leaflet signal in PC liposomes containing 5.0 mol% CL. Similar effects were observed in PC liposomes containing 2.5 mol% CL treated with either CTI or CTII (data not shown). Our ^1^H-NMR data are consistent with our ^31^P-NMR studies shown in Figure 1 and Figure 2 in that CTI and CTII mediate the formation of non-bilayer structures containing immobilized phospholipids. It should be also noted that our ^1^H-NMR data show that the formation of non-bilayer structures does not compromise the barrier integrity of liposomal membranes as they remain impermeable to Fe(CN)_6_^3−^ ions. Secondly, our ^1^H-NMR spectroscopy data indicate that PC molecules are involved in the formation of non-bilayer structures containing immobilized phospholipids. Next, we estimated the amount of non-bilayer structures containing immobilized PC by calculating the area under the high-field peak signal of N^+^(CH_3_)_3_ from PC molecules located on the outer leaflet of liposomes (see arrow in Figure 3B,C). By using this technique, we calculated that one molecule of CTII binds to four PC molecules in PC liposomes enriched with 2.5 or 5.0 mol% of CL, and one molecule of CTI binds to two PC molecules in liposomes of the same composition (Table 1).

### 2.4. Computational Modeling Identified Key Residues that Mediate Long-Distance and Short-Distance Interactions of CTI and CTII with the OMM

To elucidate the molecular mechanisms by which CTI and CTII can associate with the OMM, we employed molecular dynamics (MD) simulations to study the interactions of CTI and CTII with an in silico-generated OMM. It is worth noting that solution NMR structures may contain structural inaccuracies including hard-to-solve regions such as solvent-exposed loops that contain high flexibility, and unrealistic structural and energy calculations which can confound the interpretation of molecular dynamic simulations [15]. Therefore, in order to enhance the accuracy of the molecular dynamic simulations, we optimized the top ranked solution NMR structures for CTI and CTII respectively by using homology modeling and energy minimization procedures. Overall, the optimized models yielded QMean scores which were higher compared with their respective solution NMR structures: QMean scores of 0.560 and 0.680 for top ranked solution NMR structures of CTI (PDB entry 1ZAD) and CTII (PDB entry 1CB9), respectively, vs. 0.638 and 0.712 compared for their respective optimized models. Therefore, we used these optimized solution NMR structures to study the interaction of CTI and CTII within the OMM by molecular dynamics.

The simulation system consisted of a lipid bilayer made up of 128 1-palmitoyl-2-oleoyl-phosphatidy-choline (POPC) molecules and three CL molecules embedded in the bilayer, which is similar to the phospholipid composition in OMM [16]. The time evolution analysis suggests that several polar and charged residues in CTI and CTII interact with the OMM. The trajectories of the CL–toxin interactions showed that CTI and CTII can both interact with at least one CL and a maximum of two CL in POPC+ CL membranes (Figure 4). In one representative set of 20 ns simulations, CTII was able to electrostatically contact CL three times faster (within 4 nanosecond (ns)) compared with CTI (after 12 ns) as suggested in the time evolution plots that measure the distance of the cardiotoxin to the CL molecules (Figure 4), or the distance of the cardiotoxin to the POPC molecules (Appendix A), suggesting that CTII binds the OMM with higher avidity. Similar molecular dynamics results were observed for the top ranked solution NMR structures of CTII in that CTII binds to the OMM within 4 ns (data not shown). The MD simulation structures at 20 ns of the toxin–bilayer complex (Figure 5A) show that CTII and CTI interact with the membrane surface in multiple orientations. In two representative orientations, a series of snapshots from the simulation trajectory showing details of the toxin–CL interactions (Figure 5B) suggest that CTII interacts with the membrane surface at 10 ns via hydrophilic and hydrophobic attractions mediated by R36 and L9 residues in which CTII adapts a horizontal orientation of the long toxin axis along the membrane surface. At 20 ns, the L9 residue of CTII penetrates a bit deeper into the hydrophobic membrane region while R36 remains attracted in the polar membrane region (Figure 5(Ba)). The initial binding of CTI to the membrane surface at 10 ns is mediated by the electrostatic attraction of K18 to the polar head groups of the CL molecule in which CTI also acquires a horizontal orientation along the membrane surface (Figure 5(Bb)). However, at 20 ns, CTI changes its binding orientation along the membrane surface from horizontal to vertical with the second and third loops of CTI inserted into the membrane with R36 somewhat submerged into the polar membrane region and K35 and K44 just being embedded at the membrane surface (Figure 5(Bb)). Other representative orientations by which CTI and CTII bind to simulated OMM (POPC + Cl) are shown in Appendix A. Briefly, in two additional orientations, the CTII binding to the simulated membrane is still mediated by R36 and L9, while binding of CTI to the simulated membrane in one case is mediated by K44 and in another case by R36. At the end of the 50 ns MD runs, CTII was observed to penetrate the simulated lipid bilayer more (one out of two MD runs) compared with CTI (zero of two MD runs). Overall, the key amino acid residues mediating the initial interaction of cardiotoxins with OMM appear to be R36 and L9 in CTII (Figure 5(Ba)) and K18 at 10 ns, R36, K35 and K44 at 20 ns and either K44 or K35 at 50 ns in CTI (see Figure 5(Bb) and Appendix A). In the absence of CL, CTI and CTII failed to associate with the in silico lipid membranes composed exclusively of POPC in two independent 20 ns runs (Figure 6). No penetration of loop tips into the pure POPC membranes was also found for the *Naja oxiana* cardiotoxin by the steered molecular dynamics simulation [17]. Extended simulations of 50 ns of the two toxins both with and without CL, showed similar results as observed for the 20 ns simulations (Appendix A), corroborating the early events of toxin–bilayer interactions. For simulations with CL, both the toxin molecules interacted with one or two CL within 20 ns and remained associated for up to 50 ns (Appendix A). In the simulations without CL, the CTI molecule was unable to interact with the bilayer, and CTII only interacted weakly after 35 ns (Appendix A). The average number of interactions of each cardiotoxin with a CL within 20 ns are 22 +/−12 and 56 +/−19 for CTI and CTII, respectively. This finding along with our NMR data in this study suggest that cardiotoxins require CL molecules for initial binding to OMM.

In addition, we employed four different metrics in order to compare the ability of each cardiotoxin to approach the in silico OMM bilayer across all MD simulations (Appendix A). CTII appears to approach and bind the OMM bilayer faster and more avidly than CTI. The metrics include the average number of contacts that each cardiotoxin makes with the CL molecules within 20 ns, the average number of CL molecules each cardiotoxin makes contact with and the average time to contact, and the average affinity score (# of cardiolipins contacted divided by the average time to contact). For five simulations (20 and 50 ns MD runs), CTII makes 155.5 contacts on average vs. 66 for CTI (155.5 −/+47 SEM for CTII and 66 −/+ 29 SEM for CTI for N = 4 simulations each); CTII binds approximately 1.8 CLs vs. 1.2 for CTI for five MD simulations (1.8 −/+ 20 for CTII and 1.2 −/+ 0.37 for CTI for N = 5 simulations) and an average binding affinity score for CTII of 0.32 vs. 0.15 for CTI (0.32 −/+ 0.08 SEM for CTII vs. 0.15 −/+ 0.05 SEM).

After initial binding to OMM, cardiotoxins can embed into a deeper area of the membrane polar region via the formation of the new short-distance intermolecular bonds with the phospholipid polar heads of the OMM. Therefore, in order to identify amino acid residues on the molecular surfaces of CTII and CTI that may be critical in binding to the charged and polar groups of CL and PC, we employed the AutoDock Vina software for docking CTII and CTI with the polar heads of CL and PC without the alkyl chains. We did not use the complete molecules of CL and PC for docking because hydrophobic alkyl chains of the phospholipids are unlikely to interact with charged and polar residues on the cardiotoxins surface in the membrane polar region as previously published [18]. We analyzed the spatial arrangement of the top nine docking conformations that have the best affinity energies per docked structure. The binding affinity variations between the nine conformations were −4.8 to −4.4 kcal/mol for CTII bound to CL, −4.4 to −3.8 kcal/mol for CTII bound to PC, −5.1 to −4.7 kcal/mol for CTI bound CL, and −4.7 to −4.3 kcal/mol for CTI bound to PC. The higher amount of energy released when CL is bound to CTII and to CTI compared with when PC is bound to CTII and to CTI seems to agree with our MD data supporting a notion that CL binds to cardiotoxins with higher avidity than PC even at the level of short-range interactions. The higher binding affinities of both CL and PC to the molecular surface of CTI than to that of CTII suggest that when CTI is embedded into the membrane polar region, CTI binds anionic and zwitterionic phospholipids with the stronger force than CTII does, which may have important implications for the differences in the physiological actions of CTI and CTII.

We identified the binding sites for CL and PC that do not overlap on the molecular surfaces of CTII or CTI (Table 2). In brief, we identified three cardiotoxin binding sites of the CL and PC polar heads that share the same residue K35 on the molecular surface of CTII (Table 2). However, CL and PC make intermolecular bonds to spatially distant groups of K35 such as -N^+^H_3_ for CL and -NH^δ^^+^ of the peptide bond in K35 for PC, suggesting that both polar heads occupy different spaces on the molecular surface of CTII. The binding site in CTI employs both K35 and C38 for binding to either CL or PC (Table 2). Given that polar heads from both CL and PC make bonds to spatially distant chemical groups of K35 and C38, both polar heads do not share the same space on the molecular surface of CTI. For the presumed binding site #3 for CL and presumed binding site #2 for PC in CTI, both phospholipid polar heads interact with residues Y22 and C38 (Table 2) and make the same bonds to Y22 and C38 in CTI. Thus, the presumed binding site #3 for CL and presumed binding site #2 for PC highly overlap in a molecular space and thereby likely represent a common lipid binding site for CTI. Overall, we identified seven binding sites on the molecular surface of CTII, which include four presumed binding sites for CL and three presumed binding sites for PC. For CTI, Autodock runs identified four presumed binding sites, which include three presumed binding sites for CL and one presumed biding site for PC.

Figure 7, Figure 8, Figure 9 and Figure 10 show the binding sites with the highest binding affinity energies on surfaces of CTII and CTI for CL and PC. In brief, Autodock diagrams, in which atoms of interacting residues and polar heads are represented in a ball and stick fashion, allow a better overall 3D visualization of the intermolecular interactions, while Pymol diagrams, which utilize a stick representation for covalent bonds, facilitate the visualization of more details in the intermolecular interactions. Autodock data shown in Figure 7 and Figure 8 suggest a horizontal orientation of CTII with respect to the phospholipid membrane surface when interacting with PC + CL which is consistent with molecular dynamics (MD) simulations (Figure 5A,(Ba)). A key residue that enables the binding of CTII to CL is R36, which also agrees with the MD simulations (Figure 5(Ba)). Furthermore, Autodock data in Figure 7 and Figure 8 show the residue L9 in the binding CTII, which is involved in the initial binding to OMM according to the MD simulations (Figure 5(Ba)). However, our AutoDock docking did not identify any hydrogen bonds formed by L9 that could facilitate short-distance interactions with CTII. In addition, Autodock data suggest that residues K5, K12, K18 and K35 in CTII are solvent-exposed to mediate long-range electrostatic interactions with anionic polar heads [19] of neighboring membranes.

Figure 9 and Figure 10 demonstrate a vertical orientation of CTI in biding the surface of a phospholipid membrane consisting of PC + CL with the second and third loops initially immersed into the polar membrane region as suggested by the MD simulation studies (Figure 5(Bb) at 20 ns). K18 along with K35, R36 and C38 in CTI were identified as the key amino acid residues that bind to CL which is consistent with our MD simulations. K35, R36 and C38 in CTI were identified as the key amino acid residues that bind to PC. In addition, K2, K5, K12 and K44 are exposed away from the molecular surface of CTI, suggesting that these amino acid residues may mediate long-distance electrostatic interactions with the anionic polar heads from phospholipids in neighboring membranes [19] to promote the surrounding of the CTI molecular surface with the polar heads of phospholipids.

## 3. Discussion

### 3.1. A Converging Mechanism by Which Snake Venom Cardiotoxins Bind to the OMM

Cardiotoxins isolated from cobra venom, also known as cytotoxins, have been the subject of intense investigation for approximately five decades [20]. Snake venom cardiotoxins have elicited a high level of attention by biochemists, molecular biologists and pathologists due to their strong hemolytic activity [21], which was once believed to be one of the major factors that drive the pathology induced by envenomation from a cobra bite. Cardiotoxins attack cellular membranes causing cell lysis and numerous reports suggest the high efficiency of cardiotoxins in killing cancer cells [1,14,22,23,24]. Given the various pathological and biomedical applications of cardiotoxins, a high level of interest has been emphasized in elucidating the molecular mechanisms by which cardiotoxins interact with lipid bilayers [1,25]. Over the years, there have been numerous mechanisms proposed to explain how cardiotoxins elicit their pathological/physiological effects in affected tissues including the formation of membrane pores (also characterized as channels of specific and non-specific permeability), via direct interaction with membrane proteins, and by disrupting the organization and packing of lipids in the phospholipid membranes of cells [1,25,26]. Additional research suggested that cardiotoxins can target and bind to different organelles including lysosomes [23] and mitochondria [27,28]. Over the course of three decades, we have shown that cardiotoxins induce dehydration of the membrane surface [1,11,25], aggregation of neighboring membranes [9], intermembrane exchange of phospholipids [1,9,10,25], isotropic distribution of membrane phospholipids [29], increased membrane permeability [1,6,9], modulation of PLA_2_ [30,31] and mitochondrial ATP-synthase activities [3,4,5,25], and membrane fusion through two distinct molecular mechanisms: via formation of non-bilayer structures [10,11] and via an asymmetric enlargement of the monolayer surfaces [32]. In our recent studies, we have been focused on analyzing the effects of cardiotoxins on mitochondrial membrane structure and function [1,3,4,5,6]. Overall, we have shown that cardiotoxins CTII and CTI induce the formation of aberrant non-bilayer structures in mitochondrial membranes which presumably disrupts the normal physiological functioning of mitochondrial membranes [6]. In model membranes that mimic the phospholipid composition of mitochondrial membranes, we showed that treating model membranes at high concentrations of CTII and CTI, apart from generating non-bilayer structures, cardiotoxins promote the permeabilization, dehydration and fusion of large unilamellar membranes comprised of phosphatidylcholine and anionic phospholipids, suggesting that membrane dehydration and fusion are important steps in mediating the translocation of cardiotoxins to the inner mitochondrial membrane [6]. However, at low concentrations, ^31^P-NMR studies revealed that CTII can elicit the formation of non-bilayer structures containing immobilized phospholipids in mitochondrial membranes, a phenomenon that is accompanied with an increase in ATP-synthase activity [3,4,5]. It is worth noting that the ^31^P-NMR signal derived from non-bilayer immobilized phospholipids also exists in mitochondria at physiological conditions (in the absence of CTII) which can be elevated by increasing the temperature or lowering the pH in mitochondria [3,4,5]. The same ^31^P-NMR signal was observed in multilayer liposomes composed of PC + 20% mol CL and containing the reconstituted subunit of the F_0_-membrane of ATP-synthase [3,4,5]. These data suggest that non-bilayer immobilized phospholipids play an important physiological role in modulating mitochondrial ATP-synthase activity. In a very recent study, we have shown that another cardiotoxin, VII4, from *Naja mossambica* cobra rapidly translocates to mitochondria in cortical neurons and in neuroblastoma cells to promote aberrant mitochondrial fragmentation, a decline in oxidative phosphorylation and decreased energy production [6]. Cardiotoxin VII4 binds specifically to CL in PC membranes enriched with CL to promote non-bilayer immobilized phospholipid structures which are presumably needed for the cardiotoxin to penetrate the mitochondrial membranes [6].

By employing ^31^P-NMR spectroscopy in this study, we show for the first time that CTI elicited the formation of non-bilayer immobilized phospholipids, a molecular event that is associated with an increased ATP-synthase activity. Therefore, both CTI and CTII can modulate ATP-synthase activity in isolated mitochondria. Although we have previously demonstrated that CTII produced similar effects on mitochondria as CTI, we ran ^31^P-NMR experiments with both cardiotoxins in order to compare their ATP-synthase and non-bilayer-inducing activities in mitochondria side-to-side. Overall, our biophysical experiments suggest that CTI is less efficient in eliciting the formation of non-bilayer immobilized phospholipids and in stimulating ATP-synthase activity compared with CTII. Once we observed that CTI can modulate mitochondrial ATP synthesis and the formation of non-bilayer immobilized structures in isolated mitochondria, albeit less efficiently than CTII, we focused on analyzing the effects of both cardiotoxins on model PC membranes containing a small amount of CL in order to elucidate how CTII and CTI interact and translocate in the OMM. In brief, our, ^31^P-NMR spectroscopy studies in multilamellar liposomes comprised of PC + CL demonstrated that CTII induces the formation of non-bilayer immobilized phospholipids more efficiently in liposomes containing 2.5 and 5.0 mol% CL relative to CTI. In fact, one mole of CTII immobilized eight moles of phospholipids in multilamellar liposomes comprised of 5.0 mol% CL and six moles of phospholipids in 2.5 mol% CL, while the numbers of moles of immobilized phospholipids by one mole CTI were five and four, respectively. However, it is worth noting that other ampipathic proteins, including the purified F_0_ subunit of ATP synthase or targeting the subunit of phospholipase A2, cannot interact with mitochondria, are unable to form non-bilayer structures or modulate ATP synthase activity, suggesting that CTI and CTII specifically interact with mitochondria in in vitro systems. In live cells, another cardiotoxin (cardiotoxin IIb) but not a control reagent (Rhodamine conjugate) was able to specifically translocate to mitochondria [1,3,4,5,6].

We utilized ^1^H-NMR spectroscopy of unilamellar liposomes in a solution of Fe(CN)_6_^3^^−^ in order to determine the extent to which PC molecules are involved in the formation of non-bilayer immobilized phospholipids. In brief, we observed that one mole of CTI immobilized two moles of PC in PC liposomes containing either 2.5 or 5.0 mol% CL, while one mole of CTII immobilized four moles of PC in the same liposomes. By comparing the number of moles of immobilized PC with the number of moles of total immobilized phospholipids, we can conclude that one molecule of CTII immobilizes four CL + four PC molecules in membranes containing PC + 5.0 mol% CL, and two CL + four PC molecules in membranes containing PC + 2.5 mol%. In addition, one molecule of CTI immobilizes three CL + two PC in membranes containing PC + 5.0 mol% CL and two CL + two PC in membranes containing PC + 2.5 mol% CL. In all membrane systems employed in this study, we used a molar ratio of phospholipids and cardiotoxins at 80 to 1. This means that there are 4 CL molecules and 76 PC molecules for one cardiotoxin molecule in PC + 5.0 mol% CL membranes and 2 CL molecules and 78 PC molecules for one cardiotoxin molecule in PC + 2.5 mol% CL membranes. Thus, we can conclude that CTII immobilizes all of the four CL molecules available in PC + 5.0 mol% CL membranes, while CTI immobilizes three out of four CL molecules available in PC + 5.0 mol% CL membranes. In PC + 2.5 mol% CL membranes, both cardiotoxins each immobilize two CL molecules, which is the maximum number of CL molecules available in 2.5 mol% CL membranes for each cardiotoxin molecule. Overall, these findings are consistent with our MD data that suggest that at the initial stage of CTII and CTI interaction with the surface of a virtual membrane consisting of 128 POPC and 3 CL (POPC + 2.29 mol% CL at a phospholipid to cardiotoxin molar ratio 131 to 1), cardiotoxins bind to two molecules or one molecule of CL on the surface of the virtual membrane.

In addition, our molecular docking analysis in this study revealed that CTII harbors up to seven potential lipid binding sites that include biding sites for four CL and three PC molecules, while CTI harbors four binding sites that include binding sites for three CL and one PC molecules. The number of lipid binding sites in CTI and CTII that interact with PC molecules, as deduced by the docking analysis, are somewhat less than the number of binding sites for PC molecules as determined by ^1^H-NMR spectroscopy. This small discrepancy could be explained by the fact that AutoDock Vina, the most advanced docking program currently available, can dock large molecules like cardiotoxins with only rigid non-rotatable bonds, while cardiotoxins at physiological conditions are dynamic molecules with rotatable bonds which are probably capable of binding slightly more PC molecules than the virtual cardiotoxins we used in our docking study.

### 3.2. Proposed Molecular Mechanism by Which CTI and CTII Interact with OMM and Penetrate the OMM

According to our collective biophysical and in silico data, R36 is a key residue on the molecular surface of CTII and K18 is the key residue on the molecular surface of CTI that drive the initial electrostatic attraction of cardiotoxins to the CL polar heads on the OMM surface. In one representative orientation, CTII settles down on the OMM surface in a horizontal orientation in which the long axis of CTII partially embeds on the membrane interface, while CTI approaches the OMM surface in a vertical orientation in which the long axis of CTI embeds the membrane surface in a way that allows the second and third loops of the N-terminal region of CTI to submerge into the membrane. It has been previously noticed that the hydrophobic amino acid residues of cardiotoxins from cobra species are found only in the three loops of cardiotoxins (Figure 11) [25]. This observation gave rise to believe that the three loops of cardiotoxins penetrate deep in the hydrophobic region of membranes, a concept that has not been proven experimentally [25]. In our view, the penetration of hydrophobic loops of cardiotoxins through the polar lipid head membrane region is not a thermodynamically favored event. It should be noted that the second loop of CTI is significantly more hydrophilic than the second loop of CTII (62% of hydrophilic residues in the second loop of CTI as opposed to 38% of hydrophilic residues in the second loop of CTII—see Figure 11), suggesting that CTI has greater preponderancy of penetrating the membrane polar region with its second loop than CTII. Thus, our MD data (20 and 50 ns runs) showing the embedding of CTII on the membrane surface predominantly in a horizontal orientation and CTI penetrating the membrane polar region predominantly with its second loop, which may also involve its third loop, are consistent with the distribution of hydrophobic and hydrophilic residues within the loops of CTI and CTII (Figure 11).

Once cardiotoxins settle down on the OMM surface, both cardiotoxins establish multiple ionic, ion-dipole and hydrogen bonds with the polar head groups of CL and PC to facilitate further penetration of CTI and CTII into the polar region of the OMM. In should be noted that amino acid residues K18 in CTI and R36 in CTII which drive the initial interaction of cardiotoxins with the OMM surface, plus amino acid residues K2, K5, K12, K18, Y22, K23, K35, R36, C38, K44, and K50 (some of these residues presumably facilitate the further embedding of CTI and CTII into the OMM polar region and other residues are solved-exposed to potentially attract phospholipids of neighboring membranes), are highly conserved amino acid residues across the cardiotoxins of cobra species (Figure 11). This may imply that cardiotoxins share a converging mechanism by which cobra cardiotoxins interact and translocate through the OMM. Cardiotoxins may have their particular modes of binding to the OMM and various atomic sites of attraction, forcing further penetration of cardiotoxins into the polar region of the OMM. A particular manner by which a cardiotoxin interacts with the OMM is determined by the specific 3D surface topology of a cardiotoxin. However, we believe that cardiotoxins across cobra species share a converging mechanism which includes the initial binding of cardiotoxins to CL of the OMM and the formation of short-range ionic, ion-dipole and hydrogen bonds with CL and PC of the OMM that drive further embedding into the OMM and translocation through the OMM.

In order for cardiotoxins to translocate through the OMM, the molecular surface of cardiotoxins should be completely bound with the phospholipid polar heads. This could be accomplished by cardiotoxins attracting phospholipids of neighboring membranes to develop an intermembrane junction with a cardiotoxin in its center (Figure 12). A computer analysis in our previous study [33] has shown that the structure of an intermembrane junction, shown in Figure 12, in which phospholipids are immobilized by forces of attraction to a cardiotoxin and in which phospholipids are packed in a non-bilayer fashion with an isotropic orientation of the phosphodiester bonds, is consistent with the ^31^P-NMR signal at a resonance frequency ranging from 4.8 to 6.8 ppm observed in the multilayer liposomal samples of protein–phospholipid complexes. Furthermore, our thermodynamic stability study showed that an intermembrane junction including CL, a reverse wedge phospholipid, has a life span of 0.1 to 1.0 s and may play a key role in membrane aggregation, intermembrane lipid exchange, membrane fusion and protein translocation across a membrane [9,10,11,12,33].

According to the docking analysis in this study, there are four basic residues, K5, K12, K18 and K35, on CTII, and one basic residue, K44, on CTI, which are exposed to the solution, and can attract anionic lipid polar heads from the neighboring regions of the phospholipid membrane. Apart from mitochondria, cardiotoxins also target intracellular lysosomes [22], thus the polar heads of phospholipids on the outer surface of lysosomes could be good candidates for being attracted by the solvent-exposed basic residues of cardiotoxins which are already embedded on the outer surface of the OMM. Polar heads of phospholipids in micellular structures, which are abundant in cellular cytosol, could be also attracted to the solvent-exposed residues in cardiotoxins which are already embedded on the surface of the OMM. A cardiotoxin-triggered attraction between the outer surfaces of a mitochondrion and a lysosome or a mitochondrion and a micelle may serve to surround a cardiotoxin with phospholipids in an intermembrane junction. In the highly concentrated mitochondrial samples in this study, intermembrane junctions could be formed between the OMMs of neighboring mitochondria. In multilamellar liposomes, intermembrane junctions could be formed between the neighboring lamellar membranes inside liposomes. An intermembrane junction is a short-lived product of a dynamic equilibrium between the aggregation and disaggregation of neighboring membranes [33]. As our previous research data strongly suggest that cardiotoxins translocate across the OMM [1,3,4,5,6], we believe that when neighboring membranes disaggregate, a cardiotoxin in the intermembrane junction is released into the IMS of a mitochondrion. At the same time, it is reasonable to assume that on a few occasions when neighboring membranes disaggregate, a cardiotoxin is released outside the mitochondrion. There are many physico-chemical factors that drive the interactions of cardiotoxins with membranes of intracellular organelles towards a position of equilibrium, which could be shifted either toward releasing a cardiotoxin into the IMS or toward releasing a cardiotoxin outside the mitochondrion. These include physico-chemical features on the molecular surface of cardiotoxins such as charge distribution, polar and hydrophobic regions and the overall molecular surface 3D architecture. We are cognizant that additional studies need to be performed in the future that employ more robust molecular dynamics runs and sophisticated docking software to allow the docking of macromolecules with rotatable bonds. Additional studies in live cells should be performed to follow the translocation of CTI and CTII with fluorescently stained mitochondria by confocal microscopy. Finally, our studies warrant performing site-directed mutagenesis of lipid binding sites in CTI and CTII that were identified by the AutoDock simulations to corroborate out in silico data and further support the converging/unifying conceptual model by which cardiotoxins target mitochondria. In addition, future biophysical studies are needed that employ more sophisticated biophysical and biochemical approaches in order to identify all physico-chemical factors that drive the translocation of cardiotoxins through the OMM.

Overall, this study offers the following molecular mechanism by which cardiotoxins translocate through the OMM. Unlike our previously published studies in which high concentrations of cardiotoxins in isolated mitochondria and model membranes were used, our ^1^H-NMR data in this study on mitochondria/model membranes treated with low concentrations of cardiotoxins show that cardiotoxins may translocate through the OMM without disrupting the barrier structure of the OMM as cardiotoxins did not increase the permeability of liposomes, which mimic the phospholipid composition of the OMM, to Fe(CN)_6_^−3^ ions. From the results of this study and based on our previously published data, we believe that once cardiotoxins are located in the IMS at low concentration, cardiotoxins favor the formation of non-bilayer immobilized phospholipids via direct interaction with CL of the IMM [3,4,5]. This biophysical phenomenon facilitates the formation of compartments in mitochondrial cristae that have a high concentration of H^+^ and an increased number of ATP-synthase enzymes [3,4,5]. An increased flow of H^+^ into the IMS from the matrix through the ATP-synthase channel accelerates the spinning of the ATP-synthase rotor to increase the rate of ATP synthesis [3,4,5]. However, high concentrations of cardiotoxins in the IMS may disrupt physiologically active structures of the IMM, a mitochondrial pathology that is presumably caused by an increase in the population of phospholipids with rapid molecular mobility and isotropic orientation of phosphodiester bonds.

## 4. Conclusions

In conclusion, this study shows that low concentrations of CTI and CTII can stimulate mitochondrial ATP-synthase activity presumably through the generation of non-bilayer immobilized phospholipids in the IMM. This study also suggests that key amino acid residues, K18 on the molecular surface of CTI and R36 on the molecular surface of CTII, drive initial electrostatic interactions of cardiotoxins with the surface of the outer mitochondrial membrane. Multiple atoms on the surfaces of cardiotoxins, CL and PC, which are presumably involved in a series of intermolecular ionic, ion-dipole and hydrogen bonding, are essential for allowing cardiotoxins to embed into the OMM. The formation of an intermembrane junction as a result of ionic attractions between the OMM embedded cardiotoxins’ basic residues exposed toward anionic lipids on a surface of intracellular organelles appears to be an important step in mediating the translocation of cardiotoxins across the OMM.

## 5. Materials and Methods

### 5.1. Molecular Reagents

Potassium ferricyanide and cardiolipin (CL) from *E. coli* and egg yolk L-α-phosphatidylcholine (PC) were obtained from SIGMA Chemical Co. (St. Louis, MO, USA) and phospholipids were purified by using HPLC columns as described [31]. To isolate cardiotoxins CTI and CTII, Central Asian cobra *Naja naja oxiana* crude venom was obtained in a lyophilized form from ToxiVen Biotech (Coimbatore, Tamil Nadu, India). Cardiotoxins CTII and CTI were isolated from 500 mg of crude venom as previously published [20] and were further purified by cation exchange HPLC as previously described [14], and the purity of cardiotoxins was corroborated by biochemical means including SDS-PAGE and isoelectric focusing as previously described [30]. All other reagents and biochemicals used in this study were purchased from SIGMA Chemical Co. (St. Louis, MO, USA).

### 5.2. Preparation of Isolated Mitochondria

In order to study the biochemical and biophysical effects of CTI and CTII on isolated mitochondria, bovine heart mitochondria were isolated as previously described using sequential centrifugation steps [34,35] but with minor modifications. In brief, the final crude mitochondrial pellet was re-suspended in 5 mL of washing medium (0.3 M mannitol, 10 mM MOPS, 1 mM EDTA and 0.1% (*w*/*v*) BSA at pH 7.4). The re-suspended mitochondria were osmotically shocked with 5 mL of 20% (*w*/*v*) sodium succinate in washing medium for 10 min. This additional step was necessary to release water-soluble, phosphate-containing molecules of non-phospholipid nature which can significantly induce background noise during the NMR studies shown in Figure 1 and Figure 2. This step was necessary to simplify the ^31^P-NMR spectra and reduce unwanted effects caused by phosphates not derived from phospholipids, thereby facilitating the biophysical studies on the effects of CTI and CTII on the organization and remodeling of lipids in mitochondrial membranes. The osmotically shocked mitochondria were washed gently a few times in 40 mL of ice-cold washing medium (0.3 M mannitol, 10 mM MOPS, 1 mM EDTA and 0.1% (*w*/*v*) BSA at pH 7.4), after which another centrifugation was performed at 11,000× *g* for 10 min in order to resuspend the mitochondrial pellet in 5 mL of washing medium to a final protein concentration of 60 mg/mL. This amount of isolated bovine mitochondria was sufficient for funning up to three independent ^31^P-NMR experiments. Resuspended mitochondria were then assessed for their oxidative phosphorylation capacity by running complex II-driven respiration and ATP synthesis assays as previously described [34,35], but containing the following minor modifications. Following suspension of mitochondria in isotonic medium (0.3 M mannitol, 10 mM MOPS, 1 mM EDTA and 0.1% (*w*/*v*) BSA at pH 7.4), a respiratory control index (RCI) to analyze the structural integrity of mitochondria following their isolation was performed as previously published [36]. For this study, we found that the RCI values were higher than 4 (range 4–5) which is consistent with mitochondria that are highly coupled (oxidative phosphorylation and ATP synthesis) as previously reported [1].

To analyze the effects of CTI and CTII on the organization and structure of mitochondrial lipid bilayers, mitochondrial samples (derived from the same stock: 1.5 mL of mitochondria in washing buffer with a protein concentration of 60 mg/mL and phospholipid concentration of 6.4 × 10^−2^ M) were subjected to different treatments in NMR studies. For this study, isolated mitochondria were treated with 8 × 10^−4^ M CTII or CTI at 18 °C. Immediately after NMR spectra were recorded, ATP content (amount of ATP residing in mitochondria, not the rate of ATP synthesis) in mitochondrial samples treated with CTII or CTI and untreated sample controls was measured by using a protocol to assess mitochondrial content as previously described [37]. It is worth noting that we added additional specificity controls to determine the extent that the reported ATP concentrations are specific for ATP synthase activity. In brief, an aliquot of the same mitochondrial samples from the same stock were treated with oligomycin (1 μM), an irreversible inhibitor of ATP synthase, prior to treatment with CTII or CTI, which completely abolished ATP synthesis (data not shown).

To calculate the absolute amount of phospholipids present in each mitochondrial sample, we analyzed the integral intensity of the ^31^P-NMR signals from the experimental mitochondrial samples and normalized to the integral intensity of the ^31^P-NMR signals obtained from multilamellar liposomes containing known phospholipid concentrations. It is worth noting that all ^31^P-NMR experimental conditions were repeated at least three times with technical replicates for each mitochondrial sample and the phospholipid concentration for each mitochondrial fraction analyzed by ^31^P-NMR was approximately 6.4 × 10^−2^ M and analyzed by ^31^P-NMR; each technical replicate for ^31^P-NMR study yielded a variance of 8% between samples. The ATP synthase activity was expressed as μmol of ATP synthesized per mg of mitochondrial proteins as calculated from the triplicate measurements from the ^31^P-NMR sample tubes. To reduce technical variability in the assay, all mitochondrial samples were derived from the same stock of isolated bovine mitochondria. It is worth noting that the sample per sample variation was observed to be <8% (±4%) and the differences in results between experiments were less than 5%.

### 5.3. Preparation of Multilamellar Liposomes

Multilamellar liposomes for ^31^P-NMR studies were prepared by hydrating a dried lipid film, as previously published [3], but containing minor modifications required for this study. Multilamellar liposomes were made of either pure PC, PC + 2.5 mol% CL or PC + 5.0 mol% CL. The total phospholipid concentration in multilamellar liposomes was 6.4 × 10^−2^ M and the concentration of CTII or CTI was 8 × 10^−4^ M.

### 5.4. Preparation of Unilamellar Liposomes

Unilamellar liposomes for ^1^H-NMR studies were prepared by drying phospholipids in chloroform via the continuous exposure of the sample to helium in a vacuum for 1.5 h in order to form a lipid film. The film was then hydrated in a ^2^H_2_O buffer containing 10 mM Tris-HCl, pH 7.4, and 0.5 mM EDTA. The lipid suspension was sonicated for 15 min in helium media at 4 °C by employing an ultrasonic disperser, USDN-1 (St. Petersburg, Russia), at a frequency of 22 kHz. Unilamellar liposomes were centrifuged at 200× *g* for 60 min to remove heavy phospholipid aggregates and then incubated in helium atmosphere for 15 h at 10 °C. Unilamellar liposomes were made of either pure PC, PC + 2.5 mol% CL or PC + 5.0 mol% CL. The total concentration of phospholipids in unilamellar liposomes was 1.4 × 10^−2^ M. Cardiotoxins CTII or CTI dissolved in a ^2^H_2_O-containing buffer were added directly into unilamellar liposomes at the concentration 1.75 × 10^−4^ M.

### 5.5. ^31^P-NMR and ^1^H-NMR Studies

^31^P-NMR spectra of isolated mitochondrial fractions or multilamellar liposomes, and ^1^H NMR spectra of unilamellar liposomes, were recorded at 18 °C by employing a Varian XL-200 spectrometer (USA) that was set with the following parameters as previously published [1,3,4] but containing the following minor modifications required for this study. For ^31^P-NMR assays, mitochondrial samples and multilamellar liposome samples for each experiment were prepared twice, and the integral intensity measurements of ^31^P-NMR signals were performed three times for each sample. The saturation of the lamellar phase high-field resonance signal was achieved by applying a DANTE pulse train as previously described [38]. It is worth noting that the technical variation between measurements was observed to be less than 8% for mitochondrial samples, while the technical variation for multilamellar liposome samples was observed to be less than 5%.

^1^H-NMR spectra that were recorded from sonicated unilamellar liposomes that contained the shift reagent, potassium ferricyanide K_3_[Fe(CN)_6_], were recorded at 15 °C at an operating frequency of 200 MHz as previously published, but containing minor modifications required for this study [1,6,9,14]. The measurements of the integral intensity of ^1^H-NMR signals from the N^+^(CH_3_)_3_ groups of PC were done in triplicate readings. The variation between the triplicates was observed to be less than 6%.

### 5.6. Molecular Dynamics

To further corroborate our biophysical data which suggested that CTI and CTII can interact with mitochondria, we performed coarse-grained molecular dynamic (MD) simulations for 20 and 50 ns as further described below to understand how CTI and CTII are initially attracted to mitochondrial membranes in a dynamic manner via long-range interactions. As a first step prior to performing the MD runs, homology models based on the solution NMR structures of CTI and CTII were built using the Swiss Homology Modeling engines [39] from the top-ranked solution NMR structures for CTI (1ZAG) and CTII (1CB9). The homology model underwent extensive energy minimization steps due to the high variability in the NMR structures as previously published [1]. The quantity and structural stability for each homology model was determined by QMEAN scores, a scoring function for protein structures based on MD simulations and other protein models as previously published [6]. The processed CTI and CTII templates were then energy-minimized using GROMACS version 4.5.1 with the 53a6 GROMOS force field parameters for proteins and lipids. The CTI and CTII were then placed into a virtual system with a lipid bilayer with and without CL molecules and solvated.

Two simulation trajectories of 20 nanoseconds (ns) and two extended simulation trajectories of 50 ns each were generated for both cardiotoxins. The system consisted of the cardiotoxin placed initially at a distance of 40 Å from the surface of the lipid bilayer made up of 1-palmitoyl 2 oleoyl phosphatidyl choline (POPC) lipids, with and without 3 CL molecules embedded in the bilayer. Both the systems for CTI and CTII with CL contained 145 POPC lipids and 3 CL molecules. The CTI system had 18,999 water molecules, while the CTII had 20,194 water molecules and 3 chloride ions for neutralization. The overall system for CTI and CTII consisted of 65,435 and 69,011 atoms, respectively. In addition, we also ran MD simulations of CTI and CTII without CL molecules in the bilayer. In these systems without CL, for CTI, there are 128 POPC molecules, 13,864 water molecules and 6 chloride ions, while for CTII, there were 128 POPC molecules, 13,871 water molecules and 9 chloride ions. The overall system for CTI and CTII was 48,849 and 48,861 atoms, respectively. Each simulation system was first energy-minimized with 1000 steps of the steepest descent, followed by a short equilibration run of 200 picoseconds. During the equilibration, the cardiotoxin backbone atoms were fixed and water and lipid molecules were allowed to relax. This was followed by 20 ns of the production run using GROMACS version 4.5.1 [40] with GROMOS 53a6 force field parameters [41] for the cardiotoxins and phospholipids. The simulations were performed under constant number, pressure, temperature (NPT) conditions, with an integration time step of 2 femtoseconds. The LINCS [42] algorithm was used to constrain all bond lengths and the particle mesh Ewald method [43] was used to compute the electrostatic interactions. The temperature was kept constant at T = 310 K by coupling to an external temperature bath with a coupling constant of 0.1 picoseconds and a pressure coupling was employed to maintain a constant pressure of 1 bar. The force field parameters for CL were generated using PRODRG [44]. The extended 50 ns runs were generated with the same systems and protocols, with GROMACS version 5.1.4 [40]. In order to analyze the ability of CTI and CTII to bind to the *in silico* lipid bilayer (POPC + CL), the ability of each protein to interact with the CLs or POPC molecules in the simulated OMM was analyzed as previously published [45]. In addition, the binding affinity score was determined for each time evolution plot by averaging the number of contacts that each cardiotoxin made divided by the time to contact for five 20 and 50 ns MD runs as shown in Appendix A. Binding affinity score = [average # of contacts made per protein with CL/time to contact].

### 5.7. Molecular Docking

In order to identify potential CL and PC binding sites in CTI and CTII, the phospholipid head groups of CL and PC were docked with the solution NMR structures of cardiotoxins by using the AutoDockVina Version 4.2 program and using PDB coordinates of CL (bovine heart oxidoreductase crystal structure bound to CL PDB ID# 1V54), and for phosphatidylcholine (structure of PITP complexed to DOPC—PDB ID# 1T27) by using a similar methodology and parameters as previously published [46], but containing the following minor modifications required for this study. The CL and PC were further edited to remove the alkyl chains using Avogadro as previously published [47], and the overall charges were checked and energy-minimized using AutoDockVina. A grid box was set up with the following dimensions: center of x = 13.277; center of y = 25.025; center of z = −0.456; length of x = 36 Å; length of y = 46 Å; and length of z = 36 Å. The setting for exhaustiveness was set up as 16, which gave us consistent results in at least three sets of docking for each ligand and cardiotoxin pair in this study. Following each Autodock run, the best nine docked conformations were analyzed for ionic, ion–polar and hydrogen bond interactions between the phospholipid polar head groups and charged and polar amino acids of cardiotoxins by using Python Molecular Viewer (MGL Tools, The Scripps Research Institute).

### 5.8. Statistics

Unless indicated otherwise, most results are expressed as mean ± S.E.M. from three independent experiments or a representative experiment. Data were analyzed by Student’s *t* test (two-tailed) for single comparisons. Multiple group comparisons were done by performing one-way ANOVA followed by Bonferroni-corrected Tukey’s test. P values less than 0.05 were considered statistically significant.

## Figures and Tables

**Figure 1 toxins-12-00425-f001:**
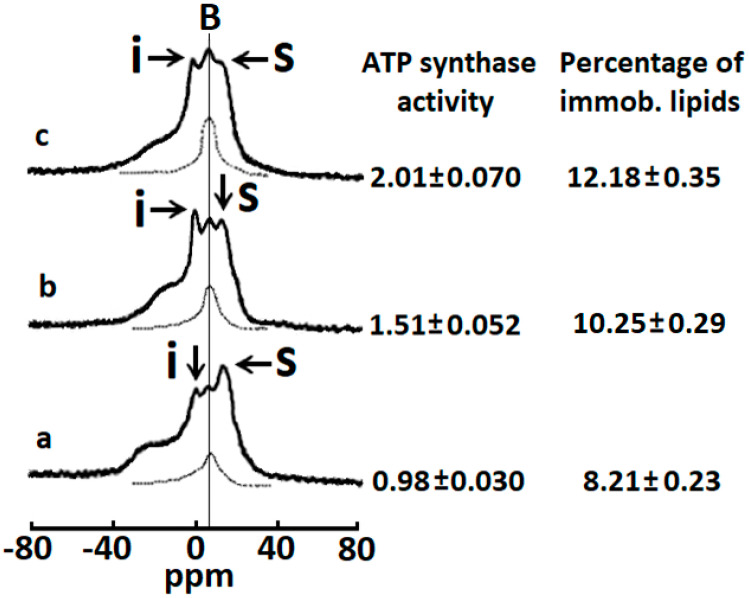
CTII and CTI modulate ATP synthase activity in mitochondria. ^31^P-NMR spectra of mitochondria recorded in the presence of 1.5 mM succinate at 18 °C in a control sample (**a**), and in the samples treated with 8 × 10^−4^ M CTI (**b**), or with 8 × 10^−4^ M CTII (**c**). The phospholipid concentration in each mitochondrial sample was estimated to be approximately 6.4 × 10^−2^ M, which was assessed by normalizing the integrated density of the ^31^P-NMR signals from the mitochondrial samples to the integrated density of the ^31^P-NMR signals from large multilamellar liposomes. Hatched lines are saturation spectra observed after applying a DANTE train of saturation pulses at the high-field peak of the lamellar spectrum (arrow with letter **S**, 11.9 ppm). The position of the hatched line signals in saturation spectra coincides with the position of ^31^P-NMR signal **B** (6.5 ppm) derived from non-bilayer immobilized phospholipids. The percentage of non-bilayer immobilized phospholipids was estimated by calculating the area under the hatched curve below the signal **B** and are shown on the column on the right as means with standard errors (± SEM) compiled from three independent experiments. The ATP levels, expressed as μmol ATP synthesized per mg of mitochondrial proteins, was monitored by taking measurements on aliquots from the ^31^P-NMR sample tubes. ATP levels are shown on the left column as means with standard errors (± SEM) compiled from three independent experiments. Statistical differences were observed between ATP-synthase activity in untreated mitochondria (**a**) vs. mitochondria treated with CTI (**b**) or CTII (**c**) (one-way ANOVA, Fisher’s Least Significant Difference). Each ^31^P-NMR spectrum shown is representative of two independent experiments that showed similar results. Each sample was measured in triplicate readings.

**Figure 2 toxins-12-00425-f002:**
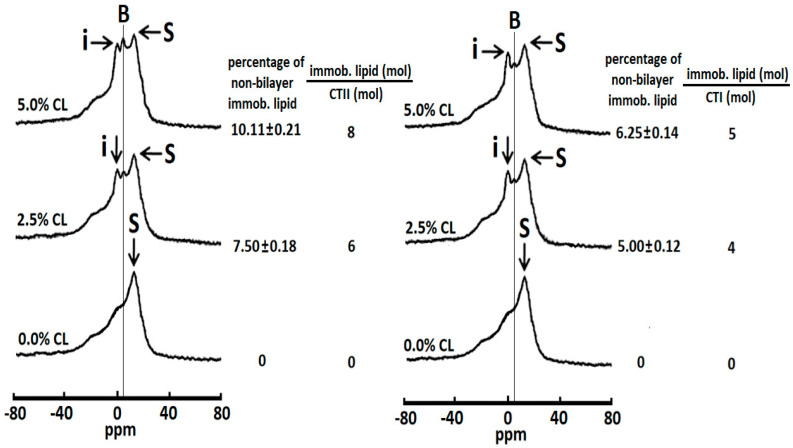
Effects of cobra cardiotoxins CTII and CTI on the ^31^P NMR-spectra of large unilamellar liposomes. ^31^P-NMR spectra of multilamellar liposomes of pure phosphatidylcholine (PC) (lower spectra), PC + 0.25% cardiolipin (CL) (middle spectra) and PC + 0.5% CL (upper spectra) containing a total phospholipid concentration of 6.4 × 10^−2^ M and treated with 8 × 10^−4^ M CTII. ^31^P-NMR spectra were recorded at 18 °C. The percentage of non-bilayer immobilized phospholipids was estimated by calculating the area under the signal **B** at 4.9 ppm remained after a DANTE train of saturation was applied at the high-field peak **S** of the lamellar spectrum (saturated spectra not shown). The percentages of non-bilayer immobilized phospholipids are shown on the column on the left as means with standard errors (± SEM) compiled from three independent experiments. Statistical differences were observed between the pure PC liposomes vs. liposomes with 2.5% CL or 5% CL (one-way ANOVA, Fisher’s LSD). Each ^31^P-NMR spectrum shown is representative of two independent experiments that showed similar results. Each sample was measured in triplicate readings.

**Figure 3 toxins-12-00425-f003:**
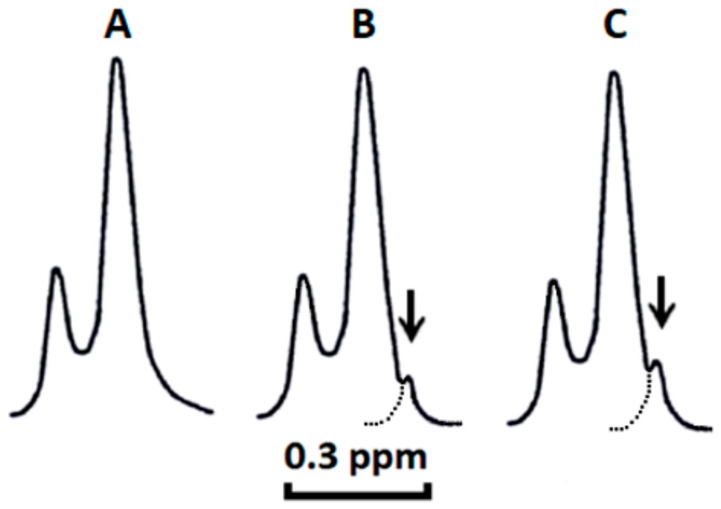
CTII and CTI induce non-bilayer structures containing PC in PC+CL liposomes, but not in PC liposomes lacking CL. ^1^H-NMR spectra derived from the N^+^(CH_3_)_3_ groups of PC in sonicated unilamellar liposomes composed of PC treated with CTII (**A**), PC+ 5mol% CL treated with CTI (**B**) and PC+ 5mol% CL treated with CTII (**C**) at a total lipid concentration 1.4 × 10^−2^ M in the presence of K_3_[Fe(CN)_6_] (10 μL of saturated K_3_Fe(CN)_6_ solution per 1 mL of liposomes) and 1.75 × 10^−4^ M CTII or CTI at 18 °C. All liposome samples were incubated for 30 min at 18 °C in a tube for ^1^H-NMR prior to adding K_3_[Fe(CN)_6_] and then incubated for another 15 min prior to adding CTII or CTI. The amount of non-bilayer immobilized PC was estimated by calculating the computer-extrapolated area under the high-field peak on the shoulder of N^+^(CH_3_)_3_ of the ^1^H-NMR signal from PC molecules on the outer leaflet of liposomes (see arrow). Each ^1^H-NMR spectrum shows representative ^1^H NMR traces from three independent experiments that showed similar results. Each sample was measured in triplicate.

**Figure 4 toxins-12-00425-f004:**
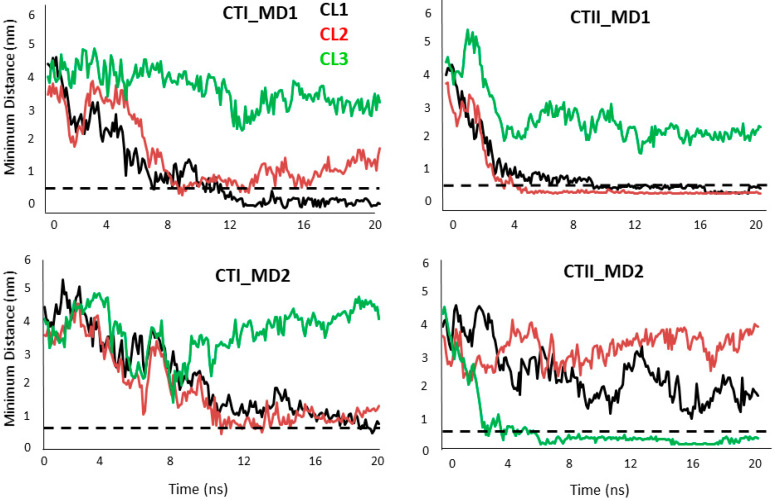
Evolution analysis of the CTI (two molecular dynamics (MD) simulations shown on left panels top and bottom) and CTII (two MD simulations shown on right panels top and bottom) interactions with the 3 cardiolipin (CL) molecules present in the bilayer. The dashed line indicates the minimum distance (0.5 nm) required for an intermolecular interaction. Note that in three of the four systems, the cardiotoxin is able to interact with at least one or two CLs within 10 ns for CTI and 4 ns for CTII and remain associated for the rest of the simulation period.

**Figure 5 toxins-12-00425-f005:**
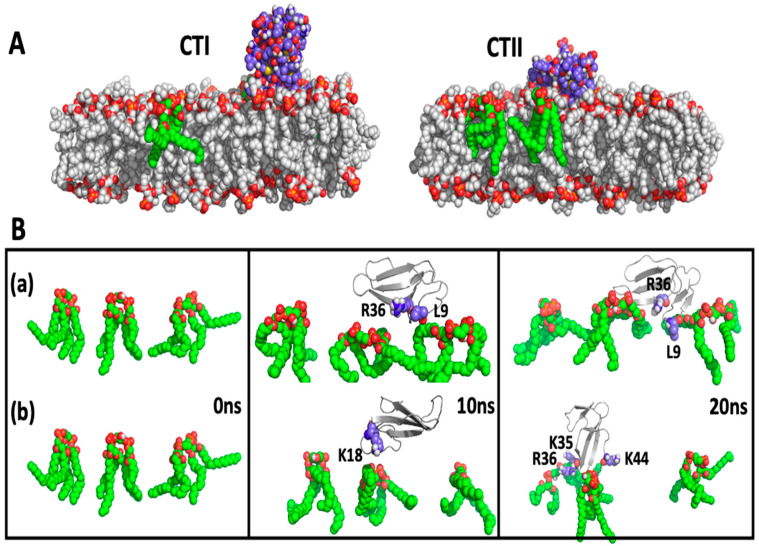
Cardiotoxins CTI and CTII interaction with the outer mitochondrial membrane (OMM). Ball and stick representations of CTI (left panel) or CTII (right panel) interacting with a lipid bilayer of POPC+CL at the end of a molecular dynamics run (20 ns) (**A**). Several representative snapshots of the interaction of CTII (a) and CTI (b) with POPC+CL membranes at 0 ns (left panel), 10 ns (middle panel) and 20 ns (right panel) time intervals (**B**). For clarity, the POPC molecules have been masked for each snapshot.

**Figure 6 toxins-12-00425-f006:**
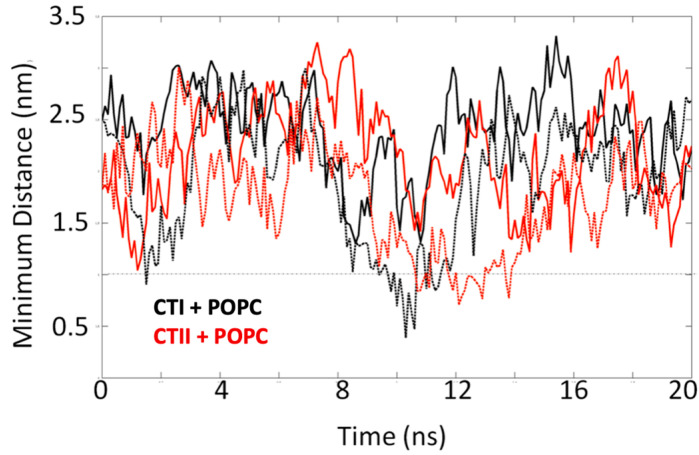
Evolution analysis of the interaction of CTI (black lines) and CTII (red lines) with simulated membranes composed exclusively of POPC for two simulations for each cardiotoxin. Note that both cardiotoxins are unable to significantly interact (come within 0.5 nm) with the POPC membranes within 20 ns.

**Figure 7 toxins-12-00425-f007:**
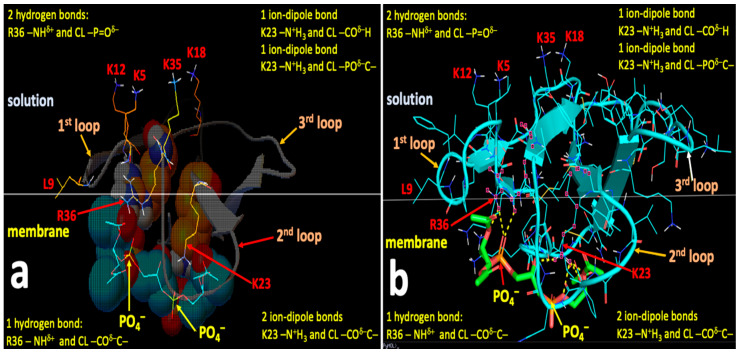
Autodock (**a**) and Pymol (**b**) diagrams showing the interaction of CTII with the polar head of CL depicted as balls-cartoon-lines (CTII) and cartoon-sticks (CL) representations. CTII binds to the phospholipid membrane surface in a horizontal orientation by employing three loop tips of CTII facing the viewer. Carbon atoms of CL are presented in the Autodock diagram as emerald balls and in the Pymol diagram as green sticks. Atoms of amino acid residues interacting with CL are shown as balls and lines (Autodock) or pink squares (Pymol). Intermolecular bonds in the Pymol diagram are shown as yellow broken lines. Types of bonds are described in yellow letter sentences both in the Autodock and Pymol diagrams. Amino acid residues K5, K12, K18 and K35, which presumably bind phospholipids of a neighboring membrane, are shown as lines both in the Autodock and Pymol diagrams.

**Figure 8 toxins-12-00425-f008:**
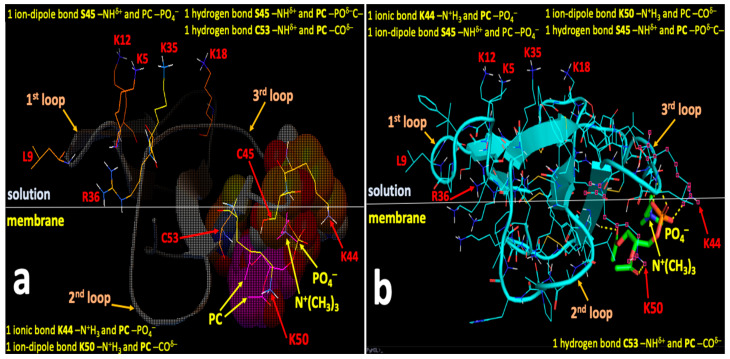
Autodock (**a**) and Pymol (**b**) diagrams showing the interaction of CTII with the polar head group of PC as shown in balls-cartoon-lines (CTII) and cartoon-sticks (PC) representations. CTII binds to the phospholipid membrane surface in a horizontal orientation by using three loop tips of CTII facing the viewer. Carbon atoms of PC are presented in the Autodock diagram as pink balls and in the Pymol diagram as green sticks. Atoms of amino acid residues interacting with PC are shown as balls and lines (Autodock) or pink squares (Pymol). The ball and stick diagram of K50 is not shown (only lines are given) as it covers the diagram of PC. Intermolecular bonds in the Pymol diagram are shown as yellow broken lines. Types of bonds are described in yellow label sentences both in the Autodock and Pymol diagrams. Amino acid residues K5, K12, K18 and K35, which presumably bind phospholipids of a neighboring phospholipid membrane, are depicted as lines in both the Autodoc and Pymol diagrams.

**Figure 9 toxins-12-00425-f009:**
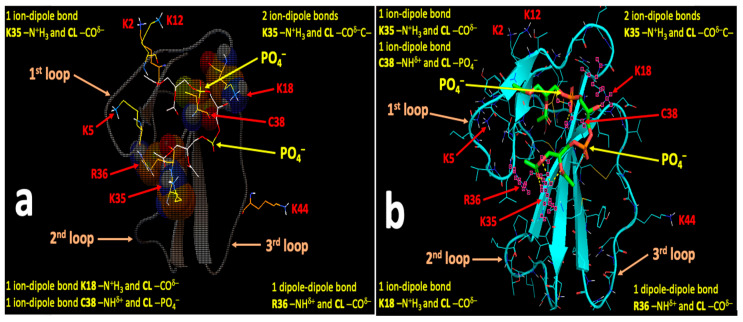
Autodock (**a**) and Pymol (**b**) diagrams showing the interaction of CTI with the polar head of CL as depicted in balls-cartoon-lines (CTI) and cartoon-sticks (CL) representations. CTI binds to the phospholipid membrane surface in a vertical orientation with its 2nd and 3rd loops involved in the initial interaction with the membrane which is followed by the complete surrounding of the CTI surface by phospholipids. The CL molecule is presented in the Autodock diagram as lines and in the Pymol diagram as green sticks. Atoms of K2, K5, K12 and K44, which are exposed away from the CTI surface and presumably support long-distance attractions to neighboring membranes, are shown in lines. Atoms of amino acid residues interacting with CL are shown as balls and lines (Autodock). In Pymol, atoms of amino acid residues interacting with CL on the front surface of CTI are shown as pink squares. Amino acid residues exposed away from the CTI surface and residues interacting with CL within CTI are identified by one letter amino acid abbreviations. Intermolecular bonds in the Pymol diagram are depicted as yellow broken lines. Types of bonds are described in yellow letter sentences both in the Autodock and Pymol diagrams.

**Figure 10 toxins-12-00425-f010:**
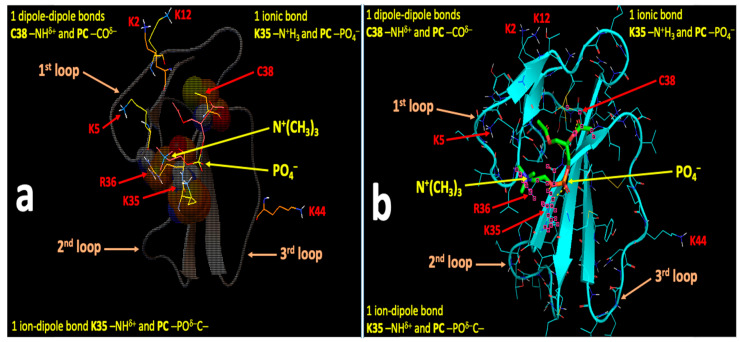
Autodock (**a**) and Pymol (**b**) diagrams show that CTI interacts with the polar head of PC as shown in balls-cartoon-lines (CTI) and cartoon-sticks (PC) representations. CTI binds to the phospholipid membrane surface in a vertical orientation with the 2nd and 3rd loops involved in the initial interaction with the membrane which is followed by the complete surrounding of the CTI surface by phospholipids. The PC molecule is presented in the Autodock diagram as lines and in the Pymol diagram as green sticks. Atoms of K2, K5, K12 and K44, which are exposed away from the CTI surface and presumably drive long-distance electrostatic interactions with neighboring membranes, are shown in lines. Atoms of amino acid residues interacting with PC are shown as balls and lines (Autodock). Amino acid residues exposed away from the CTI surface and residues interacting with PC within CTI are identified by one letter amino acid abbreviations. In Pymol, atoms of amino acid residues interacting with PC on the front surface of CTI are shown as pink squares. Intermolecular bonds in the Pymol diagram are shown as yellow broken lines. Types of bonds are described in yellow letter sentences both in the Autodock and Pymol diagrams.

**Figure 11 toxins-12-00425-f011:**
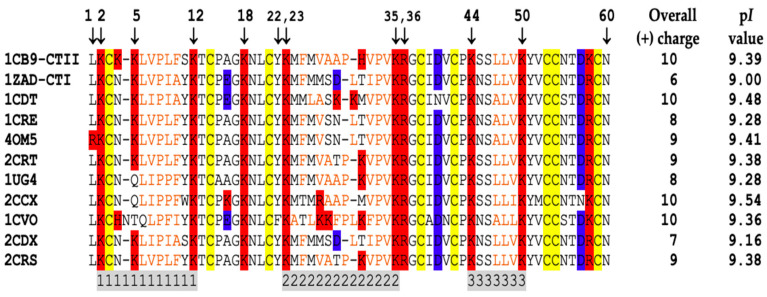
CLUSTAL 2.1 multiple sequence alignment of cobra cardiotoxins with determined 3D structures modified from our previous publication [1]. *Naja oxiana* cardiotoxins CTII and CTI along with cardiotoxins from other cobra species are indicated by their respective PDB entry codes. Conservative Cys amino acid residues shown in yellow shadows. The basic and acidic amino acid residues are shown in red and blue shadows, respectively. The amino acid residues in the three loops are indicated in grey shadows below the sequences. Hydrophobic amino acid residues of the three loops are indicated in brown color letters. The first and last amino acid residues of the sequences and predicted amino acid residues involved in the initial binding and embedding of cardiotoxins into the OMM are shown in numbers above the sequences.

**Figure 12 toxins-12-00425-f012:**
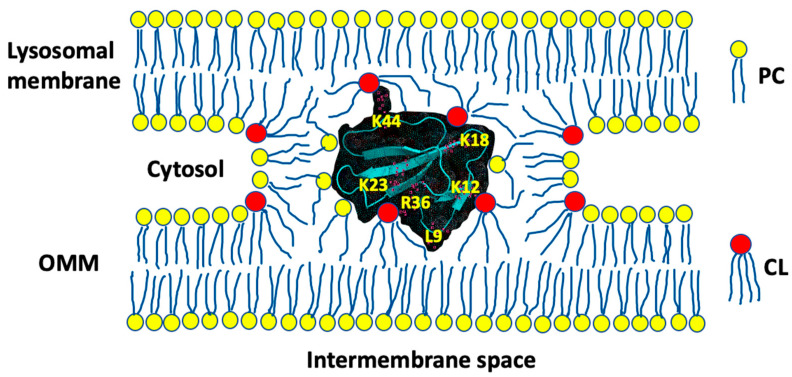
Intermembrane junction with a cardiotoxin in its center. Cardiotoxin CTII is given in a cartoon and dots representation. Amino acid residues R36 and L9 are predicted by our MD data as the key residues in the initial binding of CTII to the OMM. R36 binds to CL of the OMM while L9 penetrates to the hydrophobic region of the outer leaflet of the OMM. K5, K12, K18, K35, (not shown in this figure as they are on another side of CTII) and K44 presumably attract anionic phospholipids of neighboring lysosomes. K12 is also predicted by our AutoDock data along with K18, K23 and some other residues, which are not shown in this figure, to establish short-range intermolecular bonds with polar groups of CL and PC to support the further embedding of CTII into the OMM.

**Table 1 toxins-12-00425-t001:** Percentage of immobilized PC molecules and immobilized PC over CTI or CTII molar ratio in sonicated liposomes of pure PC, PC + 2.5% mol CL and PC + 5.0% mol CL. The amount of non-bilayer immobilized PC molecules was estimated by calculating the computer-extrapolated area under the high-field peak on the shoulder of the N^+^(CH_3_)_3_
^1^H-NMR signal from PC molecules on the outer leaflet of liposomes (see Figure 3). The percentage numbers of non-bilayer immobilized PC molecules are means with standard errors (± SEM) compiled from three independent experiments. Statistical differences were observed between pure PC liposomes vs. liposomes with 2.5% mol CL or 5% mol CL (one-way ANOVA, Fisher’s LSD).

Cardiotoxin and Liposomes System	Percentage of Non-Bilayer Immobilized PC Molecules	Immobilized PC over CTI or CTII Molar Ratio
**CTI + PC**	0	0
**CTI + PC + 2.5 mol% CL**	2.56 ± 0.14	2
**CTI + PC + 5.0 mol% CL**	2.58 ± 0.15	2
**CTII + PC**	0	0
**CTII + PC + 2.5 mol% CL**	5.26 ± 0.29	4
**CTII + PC + 5.0 mol% CL**	5.27 ± 0.29	4

**Table 2 toxins-12-00425-t002:** Cardiolipin (CL) and phosphatidylcholine (PC) binding sites on molecular surfaces of CTI and CTII determined by AutoDock computation. Amino acid residues shaded gray represent the surface area in CTI where binding sites of CL and PC overlap, thus these residues represent a common lipid binding site both for CL and PC.

	Amino Acid Residues on the Molecular Surfaces of CTII and CTI that Bind to CL and PC			
**#**	CTII + CL	CTII + PC	CTI + CL	CTI + PC
**1**	K23, R36	K44, S45, K50, C53	K18, K35, R36, C38	K35, C38
**2**	K23	K18, K35, C38	K12, K35	Y22, R36, C38
**3**	K5, K12, K35	K35	L9, Y22, C38	
**4**	K5, K18, K35			
	Total number of binding sites for CL and PC available on CTII		Total number of binding sites for CL and PC available on CTI	
	7 (4 CL + 3 PC)		4 (3 CL + 1 PC)	

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
