# Peer review of "Molecular Mechanism by Which Cobra Venom Cardiotoxins Interact with the Outer Mitochondrial Membrane"

_toxins, 2020, doi:10.3390/toxins12070425_

Round 1
Reviewer 1 Report
- What was the sequence identity (in percentage) of the homology model?
- Does the simulation system have 3 CL per monolayer? Figure 4 seems to suggest so. If so, were there any CL's on the opposite bilayer? If not, did that cause any bilayer asymmetry, and does that have an effect on the simulation results?
- The MD system description requires more details. What was the overall system size? How many waters were added? Were there any neutralizing ions?
- Lines 286-288, "CTII was able to..." is not clear and requires further evidence to make this claim. The minimum distance between three CL molecules over a 20 ns trajectory is not sufficient.
- Lines 289-291, is there any indication or past evidence available that 20 ns of simulation time is sufficient for this system to equilibrate to make observations? if there are none, then more evidence is required to support Fig. 4.
- Were the MD simulations repeated to verify reproducibility? I would like to see error-bars on Figures 4A and 4B. See recent work by Gupta et al. in JACS (https://pubs.acs.org/doi/abs/10.1021/jacs.9b13450) for example.
- Similar to above observations have been made for CTI and CTII interactions with OMM from a 20 ns trajectory. As mentioned in previous comment, perhaps running the simulations longer (e.g. 100 ns) may change a few observations.
- Figure 5 is not clear and the corresponding main text needs to explain these observations better. What is the starting position of CT1 And CTII with respect to OMM in MD simulations? As per the illustration fig. 5A, it appears that CTII has diffused in the bilayer in ~20 ns, without any bias or enhanced sampling. Is this observation true?
- How are the binding sites identified using molecular docking Autodock, validated? Also, what is the variance in binding affinity for the nine poses generated using Autodock?
- Figures 7 - 10, show the same information twice, using Autodock and Pymol rendering. The explanation provided by the authors can be found in lines 352-355. This explanation is not satisfactory. Please make single figures using any molecular visualization software (Pymol, VMD, etc..).
Other comments:
- Introduction starts with CII and then suddenly shifts to CI. Would appreciate more background on CI. How are its actions different from CII? Is there an evolutionary relationship between CI and CII?
- Figure 3 needs X-axis.
- Lines 251 - 252 says PC is involved in non bilayer immobilization. Figure 3 caption seems to suggest the complete opposite.
- Lines 72-74 say "... suggests a similar ability in enhancing ATP synthesis.... with CTI showing a decreased ability to enhance ATP synthesis". This seems a bit confusing. Does the “similar” mean that both show enhanced ATP synthesis?
- Line 134: Possible typo, looks like the authors meant Figures 1b and 1c.
- Fig 1 caption first says 1b is CTI and 1c is CTII. Later in the caption the reverse is stated.
- Discrepancy between Figure 3 caption and text. Caption says B and C are CTI and CTII with 5% CL. Text says these are both CTII with 2.5% and 5% CL.
Author Response
A note to the reviewer from the authors: Please be advised that the line numbers referred to by the reviewer does not align to line numbers in the docx file of the manuscript we submitted to the Toxins. Also, the line numbers have changed in the manuscript we revised in response to the reviewers' comments. In our answers to the reviewer’s comments we refer to the line numbers in the revised version of the manuscript (docx) we resubmitted to the Toxins.
1. What was the sequence identity (in percentage) of the homology model?-
Authors' response: Thank you for this observation, we want to clarify that the homology models that were built and optimized were based on the actual solution NMR structures for CTI (PDB:1ZAG) and CTII (PDB:1CB9). These "homology models" are actually optimized structures that were enhanced from the NMR structures by building missing small loop sequences, adding missing hydrogen bonds and optimizing amino acid side chains by using the Swiss Modeler server. Overall, the homology models used for performing AutoDock and Molecular Dynamics studies are 99% similar relative to their respective NMR structures.
2. Does the simulation system have 3 CL per monolayer? Figure 4 seems to suggest so. If so, were there any CL's on the opposite bilayer? If not, did that cause any bilayer asymmetry, and does that have an effect on the simulation results?-
Authors' response: Thank you for providing us with this observation. In order to accurately simulate a lipid bilayer of the outer mitochondrial membrane, we want to clarify that the in silico lipid bilayers (with cardiolipin, CL) only contained 3 cardiolipin molecules on one of the two monolayers of the lipid bilayer. Hence, there were no CL molecules included in the second monolayer (interior/non-solvent exposed). Since all three CLs are anionic phospholipids that actively engage with CTI and CTII via long range and medium range electrostatic interactions, we believe that placing additional CLs in the leaflet of the bilayer that does not face the cardiotoxins will not play a role in attracting CTI and CTII towards the in silico bilayer and thus not significantly change the current MD simulation results. We made this clarification on the Material/Methods section of the revised manuscript.
3. The MD system description requires more details. What was the overall system size? How many waters were added? Were there any neutralizing ions?-
Authors' response: We thank the reviewer for providing this important observation. We have included additional details regarding the size of the system (ligand and in silico POPC with/wo CL bilayers), number of water molecules and presence of neutralizing ions on the Materials/Methods section of the revised version of the research manuscript (lines 778-785)
4. Lines 286-288, "CTII was able to..." is not clear and requires further evidence to make this claim. The minimum distance between three CL molecules over a 20 ns trajectory is not sufficient.
Authors' response: We thank the reviewer for this observation. We agree that additional molecular dynamics (MD) simulations that include a longer trajectory (>20ns) is required in order to confidently show that CTII or CTI can bind the outer mitochondrial membrane by interacting with cardiolipin molecules. In response to this concern, we have now included additional MD simulations that have been run for at least 50ns showing similar results as the 20ns runs (Supplementary figure 1). Specifically, CTI and CTII can significantly interact with 1 or 2 CL molecules as evident by a robust interaction as assessed by the minimum intermolecular distance between the solution NMR structures of CTI and CTII with the CLs and the higher amount of contacts that CTII generated with the CL molecules compared to CTI (Supplementary figure 1). Hence, CTII bound to the CLs found on the outer leaflet of the in silico bilayer much avidly and faster as evidenced in our new 50ns Molecular Dynamic runs (Supplementary figure 1). Moreover, neither of the CTs interact with bilayer as efficiently when there are no CL present (Figure 5 and supplementary figure 2 with new 50ns runs). Furthermore, based on our initial biophysical data suggesting an interaction of CTI and CTII with mitochondria and micelles containing CL, we want to emphasize that premise of performing the MD simulations is to determine the extent to which CTI and CTII can initially interact with a simulated mitochondrial membrane via long range electrostatic interactions (>20Angstrons). Upon finding that CTI and CTII can interact with mitochondria per our MD simulations and collective biophysical data, we then proceeded to perform AutoDock runs in order to identify specific amino acid residues within the molecular surface of CTI and CTII that interacts with CL in the mitochondria once CTI and CTII have approached and started embedding onto the OMM. In other words, we would have not have followed up with performing AutoDock simulations if we would have found out that CTI and CTII were unable to interact with mitochondria via long range electrostatic interactions per our biophysical and MD simulations.The new data of 50ns reinforces the previous MD simulations of 20ns and further corroborates the biophysical data (NMR) that suggests that CTI and CTII can interact with micelles containing CL but not with micelles lacking CL (Figure 3, 4, 5 and supplementary figures with 50ns runs).
5. Lines 289-291, is there any indication or past evidence available that 20 ns of simulation time is sufficient for this system to equilibrate to make observations? if there are none, then more evidence is required to support Fig. 4.-
Authors' response: We thank the reviewer for this observation. Although performing 20ns MD runs may provide sufficient evidence as to whether a specific protein can interact with cardiolipin molecules in a simulated outer mitochondrial membrane, we agree that additional molecular dynamics (MD) simulations that include a longer trajectory (>20ns) is required in order to confidently show that CTII or CTI can stably electrostatically interact with the OMM over a longer time frame (>20s). In response to this concern, we have now included additional MD simulations that have been run for at least 50ns showing similar results as the 20ns runs. Specifically, CTI and CTII can significantly interact with 1 or 2 CL molecules as evident by a robust interaction as assessed by the minimum intermolecular distance between the solution NMR structures of CTI and CTII with the CLs and the higher number of contacts that CTII generated with CL compared to CTI. Furthermore, based on our initial biophysical data suggesting an interaction of CTI and CTII with mitochondria and micelles containing CL, we want to emphasize that premise of performing the MD simulations is to determine the extent to which CTI and CTII can initially interact with a simulated mitochondrial membrane via long range electrostatic interactions (>20Angstrons). Upon finding that CTI and CTII can interact with mitochondria per our MD simulations and collective biophysical data, we then proceeded to perform AutoDock runs in order to identify specific amino acid residues within the molecular surface of CTI and CTII that interacts with CL in the mitochondria once CTI and CTII have started embedding onto the OMM. In other words, we would have not have followed up with performing AutoDock simulations if we would have found out that CTI and CTII were unable to interact with mitochondria per our biophysical and MD simulations. The new data supports the previous MD simulations of 20ns and further corroborates the biophysical data (NMR) that suggests that CTI and CTII can interact with micelles containing CL but not with micelles lacking CL.
- Were the MD simulations repeated to verify reproducibility? I would like to see error-bars on Figures 4A and 4B. See recent work by Gupta et al. in JACS (https://pubs.acs.org/doi/abs/10.1021/jacs.9b13450) for example.-
Authors' response: We thank the reviewer for this observation. We want to emphasize that the previous MD runs performed for 20ns and the new 50ns simulations were performed twice for 20ns runs and twice for 50ns MD runs (total four runs) with little variability as shown by a small standard deviations across three experiments regarding the interatomic distance between the molecular surface of CTI and CTII, the number of contacts created per cardiotoxins and one CL molecule at 20ns and at 50ns. The average and standard deviations have been specified in the text (lines 319-320).
- Similar to above observations have been made for CTI and CTII interactions with OMM from a 20 ns trajectory. As mentioned in previous comment, perhaps running the simulations longer (e.g. 100 ns) may change a few observations.
Authors' Response: We thank the reviewer for raising this observation. We agree with these suggestions and have performed the suggested experiments. Please note that running new 50ns long MD simulations took five days to run on supercomputers which was barely enough time to get the data needed to address this concern and just in time to submit the revision within the deadline provided by the editors. Please see our extended response that we addressed for point #5 shown above.
8.Figure 5 is not clear and the corresponding main text needs to explain these observations better. What is the starting position of CT1 And CTII with respect to OMM in MD simulations? As per the illustration fig. 5A, it appears that CTII has diffused in the bilayer in ~20 ns, without any bias or enhanced sampling. Is this observation true?
Authors' response: We thank the reviewer for this observation. We want to clarify that the starting position used for each of the two cardiotoxins (CTI and CTII) for each of the molecular dynamics simulations was the same, specifically, approximately 40 Ang away from the in silico outer mitochondrial membrane bilayer. However, it is worth noting that the distance between the ligand (CTI or CTII) and the in silico lipid bilayer will change during the energy minimization and equilibration processes required prior to the production runs. However, at start of production runs, both CTI and CTII will still remain approximately 20-30 Ang away, which is still beyond the range of typical inter-molecular interactions. However, we want to emphasize that we have performed up to four runs that involved placing CTI and CTII at the same distance away from the lipid bilayer at different orientations which should reduce any concern that CTII for some reason ended closer to the POPC +CL bilayer following the energy minimization procedures and prior to the start of each run.
9.How are the binding sites identified using molecular docking Autodock, validated? Also, what is the variance in binding affinity for the nine poses generated using Autodock?
Authors' response: We thank the reviewer for this observation. We want to clarify that the software AutoDock predicted cardiolipin binding sites in CTI and CTII that contained the highest affinity energies (low Gibbs free energies), which are indicated in Figures 7 – 10. Our AutoDock data is consistent with the MD simulations (20ns and 50ns runs) which predict that specific basic amino acid residues in CTI and in CTII are involved in the initial interaction with cardiolipin found in the outer leaflet of the lipid bilayer of the outer mitochondrial membrane. In addition, the total numbers of lipid binding sites (cardiolipin and phosphatidylcholine) in CT and CTII as predicted by AutoDock correlate well with our biophysical data in terms of the total numbers of phospholipids immobilized by one CTI or CTII predicted by 31P-NMR and in terms of the total number of phosphatidylcholine immobilized by one CTI or CTII predicted by 1H-NMR. Please see the text in the Results section and in the Discussion that talks about this correlation. The binding affinity variations between the nine conformations generated by AutoDock were –4.8 to –4.4 kcal/mol for CTII with CL; –4.4 to –3.8 kcal/mol for CTII with PC; –5.1 to –4.7 kcal/mol for CTI with CL; and –4.7 to –4.3 kcal/mol for CTI with PC.
- A strong validation of binding sites would require studies on interaction of the point mutated recombinant CTI and CTII with liposomal membranes and isolated mitochondria.
Authors' response: We thank the reviewer for this observation. While we agree that it would be beneficial and useful to validate the lipid binding sites in CTI and CTII that are predicted to bind to cardiolipin and phosphatidylcholine (PC) by site directed mutagenesis of the recombinant versions of CTI and CTII, we want to emphasize that performing these studies will require additional resources, manpower and one or two more years of benchwork that requires expressing the recombinant forms of each toxin in insect cells using affinity purified techniques and HPLC of wild-type and over more than 10 different mutants for each cardiotoxin. Hence, we agree that the data reported in our current research manuscript warrants future studies to verify and validate the lipid binding sites for CTI and CTII but this would be beyond the scope of this paper and the very strict timeline to resubmit the first revision as imposed by the editors of the journal (1 week).
- Figures 7 - 10, show the same information twice, using Autodock and Pymol rendering. The explanation provided by the authors can be found in lines 352-355. This explanation is not satisfactory. Please make single figures using any molecular visualization software (Pymol, VMD, etc..).-
Authors' Response: We thank the reviewer for raising this observation. However, we want to clarify that the purpose for showing the molecular renderings generated by Autodock and Pymol in the same figure is to allow the audience/reader a much better enhanced visualization and appreciation of the molecular details regarding the specific amino acid residues in the molecular surface for CTI and CTII that bind to different lipids (lines 374-377) based on two different renderings. In addition, it is also worth noting that the formatting requirements for the journal Toxins does not allow the author for using double columns to publish figures in journal pages. Therefore, using two images of two versions of the renderings (Autodock and Pymol), next to each other will occupy the same space in the journal page and reduce page charges while enhancing the reader's experience and appreciation for the overall molecular structure and location of the lipid binding sites with respect to a theoretical orientation of a lipid bilayer in two types of visual renderings.
Other comments:
- Introduction starts with CII and then suddenly shifts to CI. Would appreciate more background on CI. How are its actions different from CII? Is there an evolutionary relationship between CI and CII?
Authors' response: We want to clarify that the Introduction already cites a sufficient number of papers describing extensive background research on both CTI and CTII. We want to clarify that both CTI and CTII are derived from same snake venom of Naja naja oxiana and are more than 90% homologous to each other; the comment on their homology has now been mentioned in the Introduction (lines 54-55). However, in the interest of maintaining a succinct Introduction section, we believe that describing the evolutionary relationship between CTI and CTII has little conceptual relevance and want to keep the Introduction brief.
- Figure 3 needs X-axis.
Authors' Response: We thank the reviewer for this suggestion. However, we believe that Figure 3 does not need X-axis as the 1H-NMR signals from outer and inner leaflets of liposomes separated by paramagnetic ions and are therefore can be easily identified. The ppm scale is the only information needed in Figure 3 and it is given in Figure 3. For the reader who is a novice in 1H-NMR of liposomes with paramagnetic ions the detailed explanation is given in the text of the manuscript.
- Lines 251 - 252 says PC is involved in non bilayer immobilization. Figure 3 caption seems to suggest the complete opposite.
Authors' Response: We thank the reviewer for raising this concern. Figure 3 (line 247) in the original version of the manuscript (prior to revision) reads "the amount of non-bilayer immobilized PC was estimated by calculating ....." which is in line with the text of the manuscript conveying that PC is involved in non-bilayer immobilization. To increase the accuracy of conveying our message for Figure 3, the title for Figure 3 in the revised version of the manuscript now reads "CTII and CTI induce non-bilayer structures containing PC in PC+CL liposomes, but not in PC liposomes lacking CL" The words "containing PC" have been added to the Figure 3 title (lines 241-242).
- Lines 72-74 say "... suggests a similar ability in enhancing ATP synthesis.... with CTI showing a decreased ability to enhance ATP synthesis". This seems a bit confusing. Does the “similar” mean that both show enhanced ATP synthesis?
Authors' Response: We apologize for this confusion but we believe the reviewer is referring to lines 79-82: “By running parallel experiments, a comparative analysis of both cardiotoxins suggests a similar ability in enhancing ATP synthesis via complex V in isolated mitochondria, with CTI showing a decreased ability to enhance ATP synthesis …” We have now edited the sentence to read that "both cardiotoxins can increase ATP synthesis, with CTI showing a decreased ability to enhance ATP synthesis compared to CTII" so we only substituted the word "similar" which implies the ability to increase ATP synthesis to the same level to "can increase" and then make the relative comparison of the two proteins that CTII is more potent at increasing ATP synthesis.
- Line 134: Possible typo, looks like the authors meant Figures 1b and 1c.
Authors' response : Reviewer refers to line 141 that reads (Fig. 1a and c). It is a typo and it should read (Fig. 1b and c). We thank the reviewer for noticing this typo and we corrected it.
- Fig 1 caption first says 1b is CTI and 1c is CTII. Later in the caption the reverse is stated.
Authors' response : Line 123 reads CTII (b) or CTI (c). It is a typo and it should read CTI (b) or CTII (c) and we made an appropriate correction. We thank the reviewer for noticing this typo.
- Discrepancy between Figure 3 caption and text. Caption says B and C are CTI and CTII with 5% CL. Text says these are both CTII with 2.5% and 5% CL.
Authors' response: We thank the reviewer for noticing this discrepancy which occurred because the two sentences from the earlier version of the manuscript were left in the final version of the manuscript. We rectified this omission by correcting the sentences as shown in lines 253-256 of the revised version of the manuscript.
Reviewer 2 Report
The simulation and modeling efforts made in this manuscript to elucidate the mechanisms by which CTII and CTI associate with the OMM are appreciated, but they seem to fall short in a few key ways that need to be addressed:
-
The simulation setup for the four conditions (CTII+CL, CTII+noCL, CTI+CL, and CTI+noCL) are described well, but only one single MD simulation was performed for each condition and even that one simulation was very short in duration (20 ns). The studies should be performed with multiple replicates for each condition, preferably with different orientations of the CTII and CTII proteins to gather statistical confidence and rule out the possibility that the observed protein-membrane interactions are just a result of the initial simulation conditions. A clear sign that there might be such a bias is already present in the data in Figure 4 of the manuscript where the starting distances between the CL molecules (at Time = 0 ns) is already closer for CTII than CTI. This could be controlled for by using different initial orientations across replicate simulations.
-
The same replication of the simulations for the POPC-only conditions (noCL) should also be performed, but, perhaps more importantly, a common metric needs to be used to compare the CL to noCL data. In the current manuscript, the metric used to analyze the noCL conditions is minimum distance to the bilayer (POPC) as in Figure 5. However, such data is not shown for the CL conditions. Of course, that would not be a useful metric for the CL conditions, but this begs the question of what metric should be used. I would suggest plotting the minimum distance from the protein to the center of mass of the bilayer along the Z axis for all 4 systems, which could then be plotted on the same graph for direct comparison and confirmation of the point the authors are trying to make (that the noCL conditions show little interaction between CTII or CTI with the bilayer). Multiple replicates could then allow an aggregation of the last 10ns of each simulation into a boxplot for each condition, clearly showing whether there in fact is a quantitative difference between the CL and noCL interactions with the proteins.
-
While Autodock/Pymol analysis of the noCL conditions was performed (Figures 8 and 10), there was no additional analysis of the noCL MD simulation data provided in the manuscript. Therefore, the data appear to be somewhat conflicting due to this omission: the Autodock results suggest that PC does bind to locations on CTII and CTI, but the MD simulations showed no such interactions (Figure 6). Maybe reporting some comparisons of the energies involved in these interactions could help resolve this discrepancy? Are the CL interactions energetically more favorable than the PC interactions? Regardless, in some way, the manuscript needs to show that the MD (noCL conditions) agree with the PC interactions provided by Autodock (since these are the theoretical dependents) if the reader is being asked to trust that the MD (CL conditions) agree with the CL interactions provided by Autodock.
-
One part of the analysis of the Autodock results is very unclear to me, and should be described in more detail. For Figures 7, 8, 9, 10 the molecules are each presented in some putative orientation and putative embedding into the membrane (the figures show a line with “solution” above and “membrane” below). I couldn’t find any description in the manuscript as to how both the orientation and embedding were actually obtained. Lines 355-358 reference the Autodock data as providing the necessary guidance for the orientation and insertion, but it’s not clear to me how that data was used to guide the choices concerning that interpretation and final visualization (orientation and where the solution/membrane boundary should be). By my inspection of the figures, it appears that CTII might have a decent choice in these parameters, but not so much for CTI. My skepticism comes from the results reported in Figure 5 which shows orientations from the MD simulations. The CTII MD data is not in strong agreement with the orientation in Figures 7 and 8, but at least it doesn’t seem conflicting and appears to match the Autodock data in general. However, the CTI orientation in Figures 9 and 10 suggest that the first loop of CTI is embedded in the membrane, but the MD simulation appears to clearly show the first loop pointing out up into the solvent when I compare to the structures in Figures 9 and 10 (bottom right of Figure 5… is the K18 residue mislabeled then? It looks like the first loop of CTI at 20ns is in the solvent and K18 is not being labeled properly there… the small beta sheet attached to that loop seems to be clearly in the solvent instead of in the bilayer). Even if this issue is resolved, the Autodock data could easily be used to support a more lateral placement of the CTI molecule, more similar to CTII. As a result, the choice of orientation and embedding for the CTI results looks to both conflict with the MD results and the Autodock results. (This is similar to my point 3 above: the Autodock data says both CL and PC bind in similar locations on CTI, so shouldn’t the MD simulations for the CL and noCL conditions show similar orientations relative to the membrane?) Maybe there is additional data in the literature or from another source that suggests what the CTI orientation should be, and the manuscript should include or reference those data? The manuscript currently asks the reader to put a large amount of confidence in these choices of orientation for interpretation of the results without a proper justification (or at least enumerating the assumptions and potential problems with those assumptions).
-
In several places, the representations are described as “carton” when they should be “cartoon”.
Author Response
The simulation and modeling efforts made in this manuscript to elucidate the mechanisms by which CTII and CTI associate with the OMM are appreciated, but they seem to fall short in a few key ways that need to be addressed:
1. The simulation setup for the four conditions (CTII+CL, CTII+noCL, CTI+CL, and CTI+noCL) are described well, but only one single MD simulation was performed for each condition and even that one simulation was very short in duration (20 ns). The studies should be performed with multiple replicates for each condition, preferably with different orientations of the CTII and CTII proteins to gather statistical confidence and rule out the possibility that the observed protein-membrane interactions are just a result of the initial simulation conditions. A clear sign that there might be such a bias is already present in the data in Figure 4 of the manuscript where the starting distances between the CL molecules (at Time = 0 ns) is already closer for CTII than CTI. This could be controlled for by using different initial orientations across replicate simulations.-
Authors' Response: We thank the reviewer for raising this observation and providing suggestions on improving our molecular dynamic data presented in the paper. However, we want to clarify that all molecular dynamics simulations were replicated twice for each cardiotoxin (CTI and CTII) for the 20ns simulations and additional two runs of 50ns resulting in a total of four runs for each system. For the MD run shown in the previous version of our research manuscript, we agree that CTII may have started at a slightly closer distance from the POPC bilayer containing CL. However, it is worth noting that CTII started at a closer position to the lipid bilayer relative to CTI following the energy minimization and equilibration of the system which may change the starting position of each protein before the start of the MD run. In further response to this concern and as raised by reviewer #1, we have performed two more MD simulations for a longer time frame (50ns) for each cardiotoxin with and without CL. In brief, the new data shown in the supplemental data shows that CTII is able to interact to the POPC bilayer containing cardiolipin compared to CTI. A quantitative analysis of the distance required to contact the lipid bilayers show that CTII stably binds to the in silico generated POPC bilayer compared to CTI in less than 10ns. In addition, CTII is able to produce a two-fold increase in the number of contacts with CL molecules within 20ns compared to CTI as averaged for four runs. The statistical analysis and data are shown on the figure caption of the revised version of figure 4 and for supplementary figure 1.
2. The same replication of the simulations for the POPC-only conditions (noCL) should also be performed, but, perhaps more importantly, a common metric needs to be used to compare the CL to noCL data. In the current manuscript, the metric used to analyze the noCL conditions is minimum distance to the bilayer (POPC) as in Figure 5. However, such data is not shown for the CL conditions. Of course, that would not be a useful metric for the CL conditions, but this begs the question of what metric should be used. I would suggest plotting the minimum distance from the protein to the center of mass of the bilayer along the Z axis for all 4 systems, which could then be plotted on the same graph for direct comparison and confirmation of the point the authors are trying to make (that the noCL conditions show little interaction between CTII or CTI with the bilayer). Multiple replicates could then allow an aggregation of the last 10ns of each simulation into a boxplot for each condition, clearly showing whether there in fact is a quantitative difference between the CL and noCL interactions with the proteins.
Authors' response: We thank the reviewer for raising this concern. However, we want to clarify that all of the molecular dynamic simulations for both 20ns and 50ns have been performed in duplicate (total number of 4 old and new simulations) which involved placing the optimized solution NMR structure at least 40 Angstroms away and at the same position from the lipid bilayer at different orientations of CTI and CTII with respect to the in silco-generated lipid bilayer. However, since both the protein and bilayer are dynamic entities, calculating distances from the center of mass of the lipid bilayer to the ligand (CTI and CTII) is not a standard method to assess for potential lipid-protein interactions. Finally, in order to analyze the extent to which the proteins (CTI and CTII) interact with the lipid bilayer in a more consistent manner, we have provided the standard deviations and averages of the inter-atomic distances between the CTI and CTII and the in silico lipid bilayer with/without cardiolipin of the outer mitochondrial membrane (OMM) as shown in the figure legends of figure 4 and 5 at 20 ns and in the supplemental data showing the 50ns MD runs. We also provided the statistical analyses regarding the number of contacts generated by each cardiotoxin (average and standard errors). Overall, our data shows that CTI and CTII can interact with the OMM by electrostatically interacting with CL molecules as evidenced by a significant reduction in the distance between CTI and CTII to CL in POPC bilayers containing CL and with PC molecules in POPC bilayers lacking CL.
3. While Autodock/Pymol analysis of the noCL conditions was performed (Figures 8 and 10), there was no additional analysis of the noCL MD simulation data provided in the manuscript. Therefore, the data appear to be somewhat conflicting due to this omission: the Autodock results suggest that PC does bind to locations on CTII and CTI, but the MD simulations showed no such interactions (Figure 6). Maybe reporting some comparisons of the energies involved in these interactions could help resolve this discrepancy? Are the CL interactions energetically more favorable than the PC interactions? Regardless, in some way, the manuscript needs to show that the MD (noCL conditions) agree with the PC interactions provided by Autodock (since these are the theoretical dependents) if the reader is being asked to trust that the MD (CL conditions) agree with the CL interactions provided by Autodock.
Authors' response: We thank the reviewer for raising this concern. However, we want to clarify the molecular dynamics (MD) data shown in Figure 6 depicts time evolution analysis of the electrostatic interactions of cardiotoxins (CTI and CTII) with a POPC bilayer that lacks CL which shows a lack of long range and mid-range interactions of CTI and CTII with a POPC bilayer lacking CL. In addition, our MD dynamic runs suggest that CL is responsible for attracting the CTI and CTII proteins to the lipid bilayer containing CL via long range interactions. We want to clarify that premise of performing the MD simulations was to first determine the extent to which CTI and CTII can initially interact with a simulated mitochondrial membrane via long range electrostatic interactions (>20Angstrons). However, upon finding that CTI and CTII can interact with mitochondria per our MD simulations and collective biophysical data, we then proceeded to perform AutoDock runs in order to identify specific amino acid residues within the molecular surface of CTI and CTII that interacts with CL in the mitochondria once CTI and CTII have penetrated/embdded onto the OMM after 50ns of interacting with the lipid bilayer or more. The AutoDock simulations were performed to identify the molecular mechanism by which CTI and CTII can interact with lipids (CL and PC) upon gaining entry into the outer mitochondrial membrane. In other words, we would have not have followed up with performing AutoDock simulations if we would have found out that CTI and CTII were unable to interact with mitochondria per our biophysical and MD simulations.The new data supports the previous MD simulations (20ns and 50ns) and further corroborates the biophysical data (NMR) that suggests that CTI and CTII can interact with micelles containing CL but not with micelles lacking CL.
4. One part of the analysis of the Autodock results is very unclear to me, and should be described in more detail. For Figures 7, 8, 9, 10 the molecules are each presented in some putative orientation and putative embedding into the membrane (the figures show a line with “solution” above and “membrane” below). I couldn’t find any description in the manuscript as to how both the orientation and embedding were actually obtained. Lines 355-358 reference the Autodock data as providing the necessary guidance for the orientation and insertion, but it’s not clear to me how that data was used to guide the choices concerning that interpretation and final visualization (orientation and where the solution/membrane boundary should be). By my inspection of the figures, it appears that CTII might have a decent choice in these parameters, but not so much for CTI. My skepticism comes from the results reported in Figure 5 which shows orientations from the MD simulations. The CTII MD data is not in strong agreement with the orientation in Figures 7 and 8, but at least it doesn’t seem conflicting and appears to match the Autodock data in general. However, the CTI orientation in Figures 9 and 10 suggest that the first loop of CTI is embedded in the membrane, but the MD simulation appears to clearly show the first loop pointing out up into the solvent when I compare to the structures in Figures 9 and 10 (bottom right of Figure 5… is the K18 residue mislabeled then? It looks like the first loop of CTI at 20ns is in the solvent and K18 is not being labeled properly there… the small beta sheet attached to that loop seems to be clearly in the solvent instead of in the bilayer). Even if this issue is resolved, the Autodock data could easily be used to support a more lateral placement of the CTI molecule, more similar to CTII. As a result, the choice of orientation and embedding for the CTI results looks to both conflict with the MD results and the Autodock results. (This is similar to my point 3 above: the Autodock data says both CL and PC bind in similar locations on CTI, so shouldn’t the MD simulations for the CL and noCL conditions show similar orientations relative to the membrane?) Maybe there is additional data in the literature or from another source that suggests what the CTI orientation should be, and the manuscript should include or reference those data? The manuscript currently asks the reader to put a large amount of confidence in these choices of orientation for interpretation of the results without a proper justification (or at least enumerating the assumptions and potential problems with those assumptions).
Authors' Response: We thank the reviewer for raising this concern. However, we want to clarify that the putative spatial orientations for each cardiotoxin (CTI and CTII) relative to the phospholipid membrane interface as indicated in Figures 7 – 10 per our AutoDock studies is consistent with the molecular orientation suggested by MD simulation. Specifically, Figures 7-10 show the predicted CL and PC binding sites in CTI and CTII that have the highest binding affinity energies. In addition, figures 7-10 depict a theoretical orientation of the molecular surface for CTI and CTII that is visible to readers. While we agree that AutoDock studies do not provide specific information as to how CTI and CTII may face the lipid bilayer compared to molecular dynamic simulations, we want to emphasize that the orientation of the protein shown in the AutoDock data is theoretical but based on the location of the lipid binding site (CL or PC) within each cardiotoxin and the expected orientation of each phospholipid in the context of a bilayer in a way that is consistent with our MD data. For instance, Figures 9 and 10 depict the first loop of CTI is embedded in the lipid bilayer membrane in Figures 9 and 10, while the same loop is found to be facing the in silico lipid bilayer in the same vertical orientation but more solvent-exposed during the initial interactions of CTI with the lipid bilayer prior to penetrating the bilayer (Fig 5B(b) at 20ns). Based on this orientation, we want to emphasize that K18 is properly labelled in the protein core of CTI located on the extension of 2nd loop in Figure 5B(b) and in Figure 9. In Figure 5B(b) at 10ns and 20ns, K18 clearly interacts with the CL polar head which is consistent with AutoDock rendering in Figure 9. In Figure 9 and 10, CTI is significantly more submerged into the membrane than CTI in Figure 5B(b) because AutoDock rendering represents short range ionic, ion-dipole and hydrogen interactions between cardiotoxin and phospholipids (CL and PC) that take place after initial binding of cardiotoxin to the membrane surface and which drives embedding of cardiotoxins deeper into the membrane. Furthermore, the initial electrostatic interaction of cardiotoxins with the membrane surface with the following penetration of cardiotoxins into the polar region of the lipid membrane is described in detail in the first two paragraphs in the 3.2 section of the manuscript. However, we agree that the putative orientations shown in the AutoDock data are theoretical based on limited information including the location of the lipid binding sites and the expected position of the ligands (lipids) in the context of a bilayer. Hence, we have downplayed the tone of the discussion of the results in the revised version of the manuscript.
5. In several places, the representations are described as “carton” when they should be “cartoon”.
Authors' Response: We thank the reviewer for noting these typographical errors in our manuscript. In response to this concerned, we have made the appropriate editorial corrections in the revised version of the manuscript in the figure legends of figures 7-10.
Reviewer 3 Report
- General comments
In the manuscript, it was provided a conceptual model that explains a molecular mechanism by which snake venom cardiotoxins, including CTI and CTII, interact with mitochondrial membranes to alter mitochondrial membrane structure to either upregulate ATP-synthase activity or disrupt mitochondrial function. This study shows a possibility leading the development of efficient pharmaceuticals used for increasing the bioenergetics status in cells which can serve as a pharmaceutical strategy for enhancing ATP levels and reverse a decline in energy caused by aging or chronic diseases that affect cellular bioenergetic status of cells.
- Major revision
- Figure 1
- Line 108: It is recommended to revise “(arrow with letter S)” to “(arrow with letter S, 11.9 ppm)”
- Line 108-110: It is recommended to revise “--- the position of 31P-NMR signal i ---” to “--- the position of 31P-NMR signal B (6.5 ppm) ---” and insert “letter B with arrow” in figure 3, according to Fig. 1 of Ref.3) listed in the reference.
- Line 111: It is recommended to revise “--- below the signal i ---” to “--- below the signal B ---”, according to Fig. 1 of Ref.3) listed in the reference.
- Line 122~123: It is recommended to revise the sentence “A relatively narrow peak was observed with a resonance frequency at 0.0 ppm (Fig. 1a, under the arrow shown in letter (i) in the line shape, is---” to “A relatively narrow peak was observed with a resonance frequency at 0.0 ppm (Fig. 1a, under the arrow shown in letter (i) in the line shape) is ---”.
- Line 125~126: It is recommended to revise the sentence “This broad peak signal is” to “This broad peak signal B is”.
- Figure 2 and line 178
- It is recommended to revise the sentence “the area under the signal” to “the area under the signal B” and insert “letter B with arrow” in figure 2, according to Fig. 1 of Ref.3) listed in the reference.
- Line 190: It is recommended to revise the sentence “4.9 ppm (Fig. 2)” to “4.9 ppm (Fig. 2, signal B)”.
- Figure 5 and Line 315: It is recommended to revise the sentence “--- with POPC+CL membranes at 10ns time intervals” to “--- with POPC+CL membranes at 0ns (left panel), 10ns (middle panel) and 20ns (right panel) time intervals”
- Figures 7a, 8a, 9a and 10a: It is recommended to revise both lines of polar heads (PO4-) of CL and cationic amino acids (K and R) of CTI and CTII clear and thick. In addition, it is recommended to change lines of polar heads (PO4-) of phospholipids and cationic amino acids (K and R) of cardiotoxins to another color, in order to understand the interaction of polar heads (PO4-) of CL and cationic amino acids (K and R) of CTI and CTII more easily.
- Figure 11 and line 549: As K23 and K35 (listed in line 549) were also predicted amino acid residues involved in initial binding and embedding of cardiotoxins into OMM, it is recommended to add the positions of K23 and K35 with arrows in figure 11.
- Minor revision
- Line 257: It is essential to revise the sentence”—the same composition” to “—the same composition (Table 1)”.
- Line 676: Revise the sentence “--- concentration 6.4 × 10−2 M were” to “--- concentration 6.4 × 10−2 M) were”.

Author Response
- General comments
In the manuscript, it was provided a conceptual model that explains a molecular mechanism by which snake venom cardiotoxins, including CTI and CTII, interact with mitochondrial membranes to alter mitochondrial membrane structure to either upregulate ATP-synthase activity or disrupt mitochondrial function. This study shows a possibility leading the development of efficient pharmaceuticals used for increasing the bioenergetics status in cells which can serve as a pharmaceutical strategy for enhancing ATP levels and reverse a decline in energy caused by aging or chronic diseases that affect cellular bioenergetic status of cells.
- Major revision
Figure 1
A note to the reviewer from the author (R.D.): Please be advised that the line numbers referred to by the reviewer does not align to line numbers in the docx file of the manuscript we initially submitted to the Toxins. In our answers to the reviewer’s comments we refer to the line numbers for the version of the actual manuscript (docx) that was re-submitted to Toxins and accepted for review.
- Line 108: It is recommended to revise “(arrow with letter S)” to “(arrow with letter S, 11.9 ppm)”
Author's Response: We thank the reviewer for this suggestion. In response to this concern, we have revised line 115 to read (arrow with letter S, 11.9 ppm) as recommended by the reviewer.
- Line 108-110: It is recommended to revise “--- the position of 31P-NMR signal i ---” to “--- the position of 31P-NMR signal B (6.5 ppm) ---” and insert “letter B with arrow” in figure 3, according to Fig. 1 of Ref.3) listed in the reference.
Author's Response: We thank the reviewer for this suggestion. In accordance with the reviewer’s recommendation we labeled the signal at 6.5 ppm as B (see line 118). We also labeled signal at 6.5 ppm with letter B in Fig 1 as per reviewer’s recommendation.
- Line 111: It is recommended to revise “--- below the signal i ---” to “--- below the signal B ---”, according to Fig. 1 of Ref.3) listed in the reference.
Author's Response: We thank the reviewer for this suggestion. In response to this concern, we edited the wording "below the signal i" to "below the signal B" in the revised version of the manuscript.
- Line 122~123: It is recommended to revise the sentence “A relatively narrow peak was observed with a resonance frequency at 0.0 ppm (Fig. 1a, under the arrow shown in letter (i) in the line shape, is---” to “A relatively narrow peak was observed with a resonance frequency at 0.0 ppm (Fig. 1a, under the arrow shown in letter (i) in the line shape) is ---”.
Author's Response: We thank the reviewer for this suggestion. In response to this concern, we removed ‘was’ from the “A relatively narrow peak was observed ….” in line 129 to correct a grammatical error which was noticed by the reviewer. We thank the reviewer for noticing this grammatical error.
- Line 125~126: It is recommended to revise the sentence “This broad peak signal is” to “This broad peak signal B is”.
Author's Response: We thank the reviewer for this suggestion. In response to this concern, we have now labeled the broad peak signal as the broad peak signal B as per the reviewer’s recommendation as found in lines 132-133 of the revised version of the manuscript.
Figure 2 and line 178
- It is recommended to revise the sentence “the area under the signal” to “the area under the signal B” and insert “letter B with arrow” in figure 2, according to Fig. 1 of Ref.3) listed in the reference.
Author's Response: We thank the reviewer for this suggestion. In response to this concern, we revised the sentence to “the area under the signal B at 4.9 ppm remained …” according to the reviewer’s recommendation as shown in lines 185.
- Line 190: It is recommended to revise the sentence “4.9 ppm (Fig. 2)” to “4.9 ppm (Fig. 2, signal B)”.
Author's response.: In line 197 of the revised version of the manuscript, we revised the sentence to read “4.9 ppm (Fig. 2, signal B) as per reviewer’s recommendation.
Figure 5 and Line 315: It is recommended to revise the sentence “--- with POPC+CL membranes at 10ns time intervals” to “--- with POPC+CL membranes at 0ns (left panel), 10ns (middle panel) and 20ns (right panel) time intervals”
Authors' response: We thank the reviewer for making this observation. In response to this concern, we have revised the sentence to read ““--- with POPC+CL membranes at 0ns (left panel), 10ns (middle panel) and 20ns (right panel) time intervals” as located in line 334 of the revised version of the manuscript.
Figures 7a, 8a, 9a and 10a: It is recommended to revise both lines of polar heads (PO4-) of CL and cationic amino acids (K and R) of CTI and CTII clear and thick. In addition, it is recommended to change lines of polar heads (PO4-) of phospholipids and cationic amino acids (K and R) of cardiotoxins to another color, in order to understand the interaction of polar heads (PO4-) of CL and cationic amino acids (K and R) of CTI and CTII more easily.
Authors' response.: As per the reviewer’s recommendation, the new revised Figures 7, 8, 9 and 10 we regenerated now contain symbols for PO4– of CL and amino acid residues (K and R) bigger and more clear, and we also presented amino acid residues and polar groups of phospholipids in different colors.
Figure 11 and line 549: As K23 and K35 (listed in line 549) were also predicted amino acid residues involved in initial binding and embedding of cardiotoxins into OMM, it is recommended to add the positions of K23 and K35 with arrows in figure 11.
Authors' response: As per the reviewer’s recommendation, we added arrows to indicate positions for K23 and K35 in the sequence alignment shown in Figure 11.
- Minor revision
Line 257: It is essential to revise the sentence”—the same composition” to “—the same composition (Table 1)”.
Author's Response: We thank the reviewer for this suggestion. In response to this concern,we have made the editorial correction as suggested by the reviewer as shown in line 266 of the revised version of the manuscript.
Line 676: Revise the sentence “--- concentration 6.4 × 10−2 M were” to “--- concentration 6.4 × 10−2 M) were”.
Author's Response: We thank the reviewer for this suggestion. In response to this concern, we believe that the reviewer intended to indicate an editorial correction so that the revised text reads "phospholipid concentration of 6.4 x 10-2 M in line 715 the revised version of the manuscript.
Reviewer 4 Report
Reference Manuscript toxins 818644
In this manuscript the authors are reporting results of studies on how the Cardiotoxins CT I and CT II of the snake venom of Naja oxiana interact with mitochondria and disorganize the biological functions of this organelle, with emphasis on the performance of ATP Synthase and ATP synthesis. Using purified toxins, biophysical and in silico analysis, authors showed that toxins interactions with Cardiolipin and Phosphatidylcholine interfere in the organization of phospholipid bilayers, changing phospholipid bilayers organization for non-bilayered and immobilized structures. Through molecular dynamics and molecular docking, they showed the amino acid residues involved in the interactions with mitochondria membranes and discuss aspects of the mechanism of the action of these toxins. This is a very interesting work indicating the mechanism of the action of two toxins CT I and CT II found in the snake venom of Naja oxiana, generating knowledge about these molecules and opening possibilities for future applications of these toxins in energy cell biology studies and biotechnology. However, some points need to be improved before final acceptance for publication in Toxins.
General Comments
In Figure 1, experiments were well conducted, showing the actions of Toxins CT I and CT II on mitochondrial preparations, but the authors should have used as a negative control of system stability, a toxin devoid of activities on mitochondria, to be sure that the changes described in results are actually caused by the toxins CT I and CT II, and not appeared by the simple fact of adding a protein molecule, with their hydrophobic amino acids to the system, and this could alter the lipid organization of the mitochondrial membranes, or contaminants. Interesting controls would be denatured CT I or CT II Toxins for heating, for example, or other toxins purified under the same conditions to avoid criticism of contaminants.
In Figure 2, authors studied the actions of the toxins CT I and CT II, by NMR spectra, but now on Liposomes made up of PC or PC plus increasing concentrations of Cardiolipin. The data was interesting, showing activities of both toxins on these artificial systems, however, the criticisms and suggestions made for the experiments in Figure 1, also were applied here. A protein devoid of activity on mitochondria and purified under the same conditions as CT I and CT II should have been used.
I do not criticize the conclusions pointed out by the authors, but the data would be more strengthened if analysis of images by confocal microscopy, using a mitochondrial marker, and or Transmission Electron Microscopy were used to show the changes in the organizations of the mitochondrial structures pointed out. Or using mitochondrial preparations or cultured cells.
In figure 3, authors argued that treatments with both toxins CT I and CT II caused disorganization on liposomes made up of mixtures of PC and Cardiolipin, but do not act on liposomes composed solely of PC. Once again, controls were lacking as indicated for previous experiments. I also missed a discussion about the action of these toxins on the Plasma Membrane of the cells, since they are structures rich in PC and the authors point out that both toxins CT I and CT II bind to PC. How can toxins enter inside of cells to interact with mitochondria? Can Toxins cause cell cytotoxicity by disorganizing plasma membrane?
Figures 4 and 6 show results of simulations and in silico analyzes of the need for the presence of Cardiolipin together with PC for the disorganizing activities of mitochondrial membranes caused by the toxins CT I and CT II. These data are very interesting and could explain why these toxins do not act on cellular plasma membranes, since these structures do not have cardiolin. However, they are theoretical analyzes and represent in silico simulations. Did the authors have any data from practical experiments showing crystallography for example? To strengthen these exclusively theoretical data.
In Table 2 and figures 7, 8, 9 and 10 authors showed, through theoretical data and in silico analysis, the possible amino acid residues involved in the interactions of the Toxins CT I and CT II with the phospholipids PC and cardiolipin. These data are undoubtedly consistent with the hypotheses discussed above, but would be much more strengthened if the authors can produce these toxins in their recombinant forms, but with point or combined mutations and substitutions of the indicated amino acid residues, showing that these mutations would be able to block the activities described.
Author Response
General Comments
In Figure 1, experiments were well conducted, showing the actions of Toxins CT I and CT II on mitochondrial preparations, but the authors should have used as a negative control of system stability, a toxin devoid of activities on mitochondria, to be sure that the changes described in results are actually caused by the toxins CT I and CT II, and not appeared by the simple fact of adding a protein molecule, with their hydrophobic amino acids to the system, and this could alter the lipid organization of the mitochondrial membranes, or contaminants. Interesting controls would be denatured CT I or CT II Toxins for heating, for example, or other toxins purified under the same conditions to avoid criticism of contaminants. –
Authors’ Response: We thank you for this suggestions and we understand the reviewer’s concern about having a negative control for the biophysical experiments depicted in figures 1-3. However, to appease this concern, we want to emphasize that our prior published research studies on the effects of CTI, CTII and homologous cardiotoxins/cytotoxins’ on mitochondria, cell membranes and on model membranes (micelles and liposomes) which goes over 3 decades back showing very specific effects of CTI and CTII and other cardiotoxins on lipid bilayers that contain anionic phospholipids (CL and PS, PA) and the fact that the effects on membranes are caused by CTI, CTII and homologous cardiotoxins is well documented. Despite the fact that in the last 40 years of research which shows that most research groups do not use proper negative controls to study specific interactions of membrane active properties of cardiotoxins, we want to emphasize that our research group has used denatured CTI and CTII as a negative control back in our early studies in 1980s which showed that both denatured cardiotoxins do not interact with membranes and are not able to remodel or change the structure of model membranes or have any effect on structure functional activities of cell membranes. In our recent research on mitochondria we used F0 sub-unit of ATP synthase as a negative control. In a parallel studies with CTII we showed that F0 does not have any effect on structure of model membranes of PC enriched with CL or on structure of mitochondrial membranes and mitochondrial ATP-synthase activity (Gasanov, S.E.; Kim, A.A.; Dagda, R.K. Biophysics 2016; Gasanov, S.E.; Kim, A.A.; Yaguzhinsky, L.S.; Dagda, R.K. BBA 2018). Although we do recognize the reviewer's concern about running a negative control (irrelevant protein) in these studies, we do not have the time, manpower, resources and ability to conduct another series of NMR and ATP synthase experiments within a very short time frame given the editorial deadline imposed by the journal Toxins to have a resubmission of our paper ready. However, we hope that the historical negative controls (denature CTI and CTII proteins and ATP Synthase F0 proteins) can appease this concern. We have cited the relevant papers and added a discussion on the historical negative controls used prior to this study to dispel initial concerns on a lack of specificity of cardiotoxins for binding mitochondria in the context of our data presented in this paper. We have included a discussion on the existence of prior historical negative controls which provide additional credibility on the specific activity of CTI and CTII in binding to mitochondria and modulating ATP synthase activity (lines 507-512 of the revised version of the manuscript).
In Figure 2, authors studied the actions of the toxins CT I and CT II, by NMR spectra, but now on Liposomes made up of PC or PC plus increasing concentrations of Cardiolipin. The data was interesting, showing activities of both toxins on these artificial systems, however, the criticisms and suggestions made for the experiments in Figure 1, also were applied here. A protein devoid of activity on mitochondria and purified under the same conditions as CT I and CT II should have been used.
Authors’ Response: We understand the reviewer’s concern about a negative control which he/she already expressed in his/her first comment. Please refer to our prior answer to this concern above.
I do not criticize the conclusions pointed out by the authors, but the data would be more strengthened if analysis of images by confocal microscopy, using a mitochondrial marker, and or Transmission Electron Microscopy were used to show the changes in the organizations of the mitochondrial structures pointed out. Or using mitochondrial preparations or cultured cells.
Authors’ Response: We thank the reviewer for suggesting the need to have cell biological and molecular biology data to support the biophysical and in silico data presented in this paper. However, we want to emphasize that our research group is currently not properly equipped to conduct cell biological experiments to study the effects of CTI and CTII in live cells by confocal microscopy, and in fixed cells by electron microscopy and immunolabeling methods. However, in the past, we had extensive funding and collaborations to conduct some limited cell biological studies on the effects of other cardtioxoins on mitochondria in live neurons and neuronal cells. Indeed, we have recently previously established that other cartioxins can translocated to miotchondria in live cells. Specifically, cardiotoxin VII4, a toxin homologous to CTII and CTI, can bind to mitochondria in live neurons and neuronal cells as shown by colocalization of rhodamine-labeled cardiotoxin VII4 with MitoTrackerGreen FM-labeled mitochondria via confocal microscopy (Zhang, B.; Li, F.; Chen, Z.; Srivastava, I.H., Gasanoff, E.S.; Dagda, R.K. Toxins 2019). The effects of CTI and CTII on model and cell membranes and translocation of CTI and CTII through model membranes has been studied by us previously by means of NMR, EPR and spectroscopy methods (Aripov, T.F.; Gasanov, S.E. et al. Gen Physiol Biophys. 1989; Gasanov, S.E. et al. Biol Membr. 1990; Gasanov, S.E. et al. J. Membr. Biol. 1997; Gasanov, S.E. Gen. Physiol. Biophys. 1994, 1995; Gasanov, S.E., Gasanov, E.E. J Biol Phys 1994; Gasanov, S.E. et al. J. Clin. Toxicol. 2014; Gasanov, S.E. PLoS ONE 2015; Gasanov, S.E. et al, BBA 2018). The scope of the present study was to use innovative molecular dynamic techniques to elucidate the dynamic changes in lipid structure of mitochondrial and model membranes induced by CTI and CTII in order to unveil a molecular mechanism by which cardiotoxins bind, and thereby translocate to the OMM. In the present study we utilized NMR, molecular dynamics and AutoDock simulations. Transmission Electron Microscopy, which offers a static picture on the molecular organization of objects, is not suitable for our present study but it is an excellent suggestion and worth following up in another study upon having more resources and funding. We have revised our Discussion section to reflect this concern on the need to perform future studies as a follow up to this research manuscript (lines 640-645)
In figure 3, authors argued that treatments with both toxins CT I and CT II caused disorganization on liposomes made up of mixtures of PC and Cardiolipin, but do not act on liposomes composed solely of PC. Once again, controls were lacking as indicated for previous experiments.
Authors’ Response: Thank you for bringing this concern to our attention. We want to emphasize that indeed, we employed pure PC membranes in liposome systems and virtual membranes as a negative control (see Fig. 3 and Fig 6), which showed that CTI and CTII do not interact with pure PC membranes.
I also missed a discussion about the action of these toxins on the Plasma Membrane of the cells, since they are structures rich in PC and the authors point out that both toxins CT I and CT II bind to PC. How can toxins enter inside of cells to interact with mitochondria? Can Toxins cause cell cytotoxicity by disorganizing plasma membrane?
Authors’ Response: We thank the reviewer for bringing this concern. In the present study we investigated the binding of PC polar heads on the molecular surface of CTI and CTII by performing AutoDock simulations. These phenomena is expected to occur only after cardiotoxins penetrate the surface f PC+CL membrane. Cardiotoxins do not have an opportunity to bind to PC polar heads in pure PC membranes as they need CL molecules present in a lipid bilayer, or another anionic lipid (PA or PS) to gain access to the lipid bilayer via long range electrostatic interactions as shown in our 20ns and 50ns molecular dynamic simulation data (Figures 4-5 and supplemental figure 1). In addition, plasma membranes, which are rich in PC, also contain small concentration of acidic phospholipids that mediate interaction of cationic proteins with plasma membranes. The interaction of cardiotoxins/cytotoxins with plasma membranes and associated cell cytotoxicity has been documented and this model has been validated by many research groups in the area of toxinology and venemology and even cancer research (see our recent review Gasanov, S.E. et al. J. Clin. Toxicol. 2014). However, we want to emphasize that the scope of our present study was predominantly focused at understanding the dynamic changes in OMM which support translocation of cardiotoxins through OMM and build upon our prior nascent conceptual model (interaction of CTI and CTII with miotchondrial membranes) and placed a minor emphasis on understanding the ability of cardiotoxins interactions with plasma membranes for which there is more extensive knowledge on this topic.
Figures 4 and 6 show results of simulations and in silico analyzes of the need for the presence of Cardiolipin together with PC for the disorganizing activities of mitochondrial membranes caused by the toxins CT I and CT II. These data are very interesting and could explain why these toxins do not act on cellular plasma membranes, since these structures do not have cardiolin.
Authors’ Response: We thank the reviewer for this observation. Indeed, both CTI and CTII along with other cardiotoxins/cytotoxins interact with plasma membranes causing various cytotoxic effects such as cells lysis, membrane permeability, membrane fusion, depolarization of nerve cell membranes, translocation across cell membranes and etc., which is mediated by other acidic phospholipids (e.g. phosphatidic acid and phosphatydylserine), extracellular phospholipases and other factors (see our recent review Gasanov, S.E. et al. J. Clin. Toxicol. 2014). A new area/focus in research on cardiotoxins cytotoxicity was initiated based on the novel finding on the ability of cardiotoxins to interact with cardiolipin in lipid membranes from mitochondria (Gasanov, S.E. PLoS ONE 2015) which suggested another pathological mechanism by which cardiotoxins can induce cytotoxicity through the disintegration of mitochondrial membranes. Our recent research also showed that cardiotoxins, at low concentrations, do not disrupt functions of mitochondria but instead enhance mitochondrial ATP-synthase activity (Gasanov S.E. et al. Biophysics 2016; Gasanov, S.E. et al, BBA 2018) that added more interest toward understanding how cardiotoxins interact with mitochondrial membranes, which is line with the scope of our present study to understand the dynamic changes induced by cardiotoxins in mitochondria.
However, they are theoretical analyzes and represent in silico simulations. Did the authors have any data from practical experiments showing crystallography for example? To strengthen these exclusively theoretical data.
Authors’ Response: We thank the reviewer for bringing this observation to our attention. However, we want to emphasize that our collective data reported in the present study which includes in silico simulations (AutoDock and Molecular Dynamic simulations) are strongly supported by our experimental data obtained by 31P-NMR and 1H-MNR spectroscopy. Methods of crystallography do not offer any information on changes in phospholipid dynamics induced by cardiotoxins in OMM and getting static crystallographic images of mitochondrial membranes is beyond the scope of our present study.
In Table 2 and figures 7, 8, 9 and 10 authors showed, through theoretical data and in silico analysis, the possible amino acid residues involved in the interactions of the Toxins CT I and CT II with the phospholipids PC and cardiolipin. These data are undoubtedly consistent with the hypotheses discussed above, but would be much more strengthened if the authors can produce these toxins in their recombinant forms, but with point or combined mutations and substitutions of the indicated amino acid residues, showing that these mutations would be able to block the activities described.
Authors' Response: We thank the reviewer for this observation. While we agree that it would be beneficial and useful to corroborate the lipid binding sites in CTI and CTII that are predicted to bind to cardiolipin and phosphatidylcholine (PC) by site directed mutagenesis to produce recombinant mutants of CTI and CTII, we want to emphasize that performing these studies will require additional resources, manpower and one or two more years of benchwork that requires expressing the recombinant forms of each toxin in insect cells and using affinity-based purification techniques and HPLC to purify a large amount of wild-type and over more than 10 different mutants for each cardiotoxin to perform downstream biochemical assays in mitochondria as shown in our paper. Hence, we agree that the data reported in our current research manuscript warrants future studies to verify and validate the lipid binding sites in CTI and CTII but this would be beyond the scope of this paper and the very strict timeline to resubmit the first revision as imposed by the editors of the journal.
Round 2
Reviewer 1 Report
Citation to Gupta et al. in JACS is missing. Rest is good.
Author Response
We thank the reviewer for letting us know that a citation (Gupta et al., JACS) was missing in our references. In response to this concern, we have included the missing citation in the the body of the research manuscript and in the Bibliography of the revised version of the manuscript - see reference 46 - line 1042.
Reviewer 2 Report
I brought up 5 points in my last review, and here I will highlight my opinions (both the positive and negative aspects) of how the manuscript has changed in response to those points.
- The manuscript now includes two additional MD simulations (50ns) in supplementary material, and also now includes figures/data for the -other- two MD simulations (which were mentioned in the prior version's Methods section, but the data were not provided in the manuscript). Positives: I am pleased that the manuscript now includes more data comparing CTI and CTII in the various simulation conditions explored, and that all of the data is reported in some form for review by the reader. This potentially helps to address the concern that initial conditions could bias the results. These additional simulations could also help validate the specific interactions indicated by the manuscript. Negatives: The figures all report the same metric now (which is good), but all show different scales on the Y-axes which muddles comparison of the results. That should be fixed so that all figures looking at minimum distances can all be adequately compared by the reader. More importantly though, the general picture that emerges across all 4 simulations for each protein is the following: CTI doesn't appear to interact any more/less with the CL molecules than CTII. For example, across the CTI simulations 2/3 CL molecules make close contact in 3/4 of the simulations, and the other shows 1/3 CL molecules making close contact. Then for CTII, 2/3 CL molecules make close contact in 3/4 simulations, and the other shows 1/3 CL molecules making close contact. In other words, I can't see any reason to actually say that CTI interacts with CL less than CTII (or vice-versa) given these simulations. One might say that -one- of the CTI simulations with 2/3 contacts shows that one of the two is transient: but this gets back to my point that this isn't enough data to make that claim with statistical confidence. Other data in the manuscript may support this hypothesis, but the MD simulations do not.
- The manuscript (and supplementary information) now includes the figures/ data from two 50ns simulations for each of the POPC-only conditions. Positives: the additional data all show the same trends and are consistent with the hypothesis that, generally, CTI and CTII do not interact with bilayers lacking CL (note that one simulation of CTII did not show this result, but the manuscript points this out and it's not contentious like point 1 above). Negatives: These data still do not directly and concisely address the concern that the hypothesis explored in the manuscript (that POPC-only show less interactions with CTI and CTII than POPC+CL bilayers) because the comparison is supported by data provided from two different metrics. (Comparing minimum distances to individual CL molecules with minimum distances to the entire bilayer is an inconsistency that could easily be addressed.) I suggested a nonstandard metric to address this in my review in my haste, so perhaps a better approach would instead be to include a more standard common metric in the manuscript of some other kind instead of not finding a common metric at all? For example, calculating the same minimum distance to the bilayer for both POPC-only and POPC+CL simulations. This would allow for a direct comparison, and make it clear to the reader that the hypothesis is actually supported by the data. I actually anticipate the data will indeed show this to be the case, but the current comparisons do not make it clear enough for the reader. Also, yet again keeping Y-axis scales consistent between all figures with minimum distances would increase clarity for the reader.
- My earlier concern with the Autodock data was simply that it didn't help explain in more detail why the CL interactions were preferred to POPC interactions. I appreciate the perspective provided by the authors in responding to my request, but I still think that providing some of the data (Autodock scores from the predicted binding sites) would help support the hypothesis that CTI and CTII interact more strongly with CL and less strongly with POPC (as observed by the MD simulations). Positives: Since, yes, the MD simulation data was the main impetus for that hypothesis and the reason to explore Autodock, then after addressing point 2 above there should be no other evidence needed to make that claim. So, I concede this might not need to be addressed once point 2 is clarified above. Negatives: The Autodock scores are likely to correlate with the hypothesis and will probably just support and clarify this conclusion, so it seems remiss to exclude them (or some other measurement of affinity) to support it. Some readers might appreciate the confirmation, so it's a suggestion instead of a requirement.
- I appreciate the authors' attempts to aid my misunderstanding about the location of K18 on CTI and the orientation of CTI relative to the membrane, but the response itself seems inconsistent with the figures in the manuscript (just like the text of the manuscript itself). For example, the authors state the following: "we want to emphasize that K18 is properly labelled in the protein core of CTI located on the extension of 2nd loop in Figure 5B(b) and in Figure 9". However, when I look at Figure 9, I see absolutely -no- residues labeled on the 2nd loop. The second loop in that figure is actually at the top, and highly solvent exposed according to the figure. Instead, K18 is in the membrane along with the 1st loop. When looking back at Figure 5B(b) at 10ns, it indeed looks like K18 is bringing the 1st loop closely to the CL molecule. However, in the same figure at 20ns, there are -2- residues shown in spherical representations, but only one of them is labeled. The left one is labeled K18, but it looks instead like maybe those would be K44 and K35 instead? It looks like the first loop is now oriented completely away from the bilayer, which would mean that the K18 label is incorrect in Figure 5B(b) at 20ns. Also, did the other simulations show a similar result? Positives: I am fine with the idea that the Autodock data was the primary impetus for the orientations in Figures 9 and 10. Negatives: I am still concerned about the labeling on Fig 5B(b), but I can't verify it myself. It appears that the 1st loop is not in contact with the bilayer. Given the provided orientation at 20ns in Fig. 5B(b), I can't see how K18 can actually be close to the bilayer if Figures 9 and 10 are labeled correctly. More importantly, the rest of the images/data from the remaining simulations need to be reported. There are now 3 more simulations and those data need to be presented to see if they really support the putative orientations. One simulation image just isn't enough in my opinion even if everything is correct in Fig 5B(b).
- The manuscript now addresses all of my concerns on my fifth point.
Author Response
- The manuscript now includes two additional MD simulations (50ns) in supplementary material, and also now includes figures/data for the -other- two MD simulations (which were mentioned in the prior version's Methods section, but the data were not provided in the manuscript). Positives: I am pleased that the manuscript now includes more data comparing CTI and CTII in the various simulation conditions explored, and that all of the data is reported in some form for review by the reader. This potentially helps to address the concern that initial conditions could bias the results. These additional simulations could also help validate the specific interactions indicated by the manuscript. Negatives: The figures all report the same metric now (which is good), but all show different scales on the Y-axes which muddles comparison of the results. That should be fixed so that all figures looking at minimum distances can all be adequately compared by the reader. More importantly though, the general picture that emerges across all 4 simulations for each protein is the following: CTI doesn't appear to interact any more/less with the CL molecules than CTII. For example, across the CTI simulations 2/3 CL molecules make close contact in 3/4 of the simulations, and the other shows 1/3 CL molecules making close contact. Then for CTII, 2/3 CL molecules make close contact in 3/4 simulations, and the other shows 1/3 CL molecules making close contact. In other words, I can't see any reason to actually say that CTI interacts with CL less than CTII (or vice-versa) given these simulations. One might say that -one- of the CTI simulations with 2/3 contacts shows that one of the two is transient: but this gets back to my point that this isn't enough data to make that claim with statistical confidence. Other data in the manuscript may support this hypothesis, but the MD simulations do not. Authors' Response: We thank the reviewer for the insightful observations and the positive aspects of our previous revision in response to this concern. In response to the negative aspects of the new MD simulations, we want to emphasize that we have now provided new, regraphed plots of the 20 ns and 50ns simulations (figure 4 and supplementary figure 1-2) showing similar "Y" scales (distance between protein and lipid bilayer). In addition, we have provided new data (Supplementary figure 3) which includes new plots that uses a different metric: distance between the POPC molecules and CTI and CTII for both the 20ns and 50ns simulations that were performed with POPC +CL in silico bilayers. Overall, like the first set of MD simulations that measured distance from the cardiotoxin to the CL molecules, the new data shows that CTII shows a trend in approaching the phosphatidylcholine molecules in the POPC + CL bilayer more rapidly and more avidly compared to CTI. We recognize that the data shows trends and likely not sufficient statistical confidence/power to show that CTII does bind to the POPC + CL bilayer more rapidly and avidly compared to CTI. However, we want to point out that three different metrics suggest that CTII approaches the POPC + Cl bilayer more avidly compared to CTI including: the average number of contacts that each cardiotoxin makes with the CL molecules within 20ns, the average number of CL molecules each cardiotoxin makes and the average time to contact, and the average affinity score (# of cardiolipins contacted divided by the average time to contact). For instance, CTII makes 155.5 contacts on average vs. 66 for CTI (155.5 -/+47 SEM for CTII and 66 -/+ 29 SEM for CTI for N=4 simulations each); CTII binds approximately 1.8 CLs vs. 1.2 for CTI for 5 MD simulations (1.8 -/+.20 for CTII and 1.2 -/+0.37 for CTI for N=5 simulations) and an average binding affinity score for CTII of 0.32 vs. .15 for CTI (0.32 -/+ 0.08 SEM for CTII vs. 0.15 -/+0.05 SEM). We mentioned the statistical trends observed for the increased binding affinity of CTII in terms of being electrostatically attracted to POPC +CL bilayers compared to CTI. These statistical data is noe included as a table in the supplementary material and shows trends for CTII in binding the POPC + CL bilayers more avidly.
- The manuscript (and supplementary information) now includes the figures/ data from two 50ns simulations for each of the POPC-only conditions. Positives: the additional data all show the same trends and are consistent with the hypothesis that, generally, CTI and CTII do not interact with bilayers lacking CL (note that one simulation of CTII did not show this result, but the manuscript points this out and it's not contentious like point 1 above). Negatives:These data still do not directly and concisely address the concern that the hypothesis explored in the manuscript (that POPC-only show less interactions with CTI and CTII than POPC+CL bilayers) because the comparison is supported by data provided from two different metrics. (Comparing minimum distances to individual CL molecules with minimum distances to the entire bilayer is an inconsistency that could easily be addressed.) I suggested a nonstandard metric to address this in my review in my haste, so perhaps a better approach would instead be to include a more standard common metric in the manuscript of some other kind instead of not finding a common metric at all? For example, calculating the same minimum distance to the bilayer for both POPC-only and POPC+CL simulations. This would allow for a direct comparison, and make it clear to the reader that the hypothesis is actually supported by the data. I actually anticipate the data will indeed show this to be the case, but the current comparisons do not make it clear enough for the reader. Also, yet again keeping Y-axis scales consistent between all figures with minimum distances would increase clarity for the reader. Authors' Response: We agree with the reviewer with this concern. In response to the suggested changes to strengthen our MD data, we now have provided new data that employs the same metric across all MD simulations: the distance between each cardiotoxin (CTI and CTII) to the POPC molecules within the POPC + Cl bilayers. Overall, the new MD simulation plots show that CTI and CTII approach and interact with POPC + CL bilayers within less than 20 ns (most of these stable interactions happen in less than 10 ns) as measured by the distance between the cardiotoxins and the POPC molecules. Hence, by using a similar metric to analyze the results of the MD simulations done in POPC bilayers with/without CL, the new supplementary data (Supplementary figure 3) supports and corroborates the prior MD time evolution plots that measured the distance between CTI and CTII with respect to the CL molecules. In brief, the data shows that CTI and CTII interact with bilayers containing CL whereas CTI and CTII failed to significantly interact with POPC only bilayers as noted by the unstable interactions (please observe the highly variable distances between CTI and CTII with POPC bilayers vs. POPC +CL bilayers in MD simulations). Yes, it seems that in one simulation out of 4, one of the cardiotoxins interacts with POPC bilayer within 40ns but the interaction is very unstable and the CT fails to maintain enough interactions.
- My earlier concern with the Autodock data was simply that it didn't help explain in more detail why the CL interactions were preferred to POPC interactions. I appreciate the perspective provided by the authors in responding to my request, but I still think that providing some of the data (Autodock scores from the predicted binding sites) would help support the hypothesis that CTI and CTII interact more strongly with CL and less strongly with POPC (as observed by the MD simulations). Positives: Since, yes, the MD simulation data was the main impetus for that hypothesis and the reason to explore Autodock, then after addressing point 2 above there should be no other evidence needed to make that claim. So, I concede this might not need to be addressed once point 2 is clarified above. Negatives: The Autodock scores are likely to correlate with the hypothesis and will probably just support and clarify this conclusion, so it seems remiss to exclude them (or some other measurement of affinity) to support it. Some readers might appreciate the confirmation, so it's a suggestion instead of a requirement.Authors' Response: We thank the reviewer for this observation and brining this to our attention. Indeed, we now have provided the AutoDock affinity data for the binding affinity of CTI and CTII towards PC vs. CL. Overall,the affinity binding data (as described in a range of kCal/mol per lipid bound to cardiotoxin) does show that CTI and CTII bind more avidly to CL compared to PC. Authors' Response: We thank the reviewer for this observation. In response to this concern, we inadvertently forgot to provide the data on the binding affinities for each of the docked conformations in our original response to this reviewer as we were focused at addressing similar concern of another reviewer. Here, we now show that the binding affinities between the nine conformations ranged from –4.8 to –4.4 kcal/mol for CTII bound to CL, –4.4 to –3.8 kcal/mol for CTII bound to PC, –5.1 to –4.7 kcal/mol for CTI bound to CL, and –4.7 to –4.3 kcal/mol for CTI bound to PC. The higher amount of energy released when CL binds to CTII or to CTI compared when PC bound to CTII and to CTI seems to agree with our MD data which supports the concept that CL binds more avidly to cardiotoxins than to PC at the level of short range interactions. The higher binding affinities of both CL and PC to molecular surface of CTI than to that of CTII suggests that when CTI is embedded into membrane polar region, CTI binds anionic and zwitterionic phospholipids with the slightly stronger force than CTII does, which may have important implication for differences in physiological actions of CTI and CTII. We added binding affinities values and accompanying discussion into the revised manuscript in lines 351-359. However, additional data supporting the MD simulations, based on the distance of CTI and CTII to POPC molecules, does support the concept that CTII interacts with the phospholipid bilayer composed of POPC + CL more rapidly and avidly than CTI but the ability of CTI and CTII to embed the lipid bilayer by interacting with both POPC and CL is stronger for CTI. We made these clarifications in the Results and Discussion sections of the revised versions of the manuscript.
- I appreciate the authors' attempts to aid my misunderstanding about the location of K18 on CTI and the orientation of CTI relative to the membrane, but the response itself seems inconsistent with the figures in the manuscript (just like the text of the manuscript itself). For example, the authors state the following: "we want to emphasize that K18 is properly labelled in the protein core of CTI located on the extension of 2nd loop in Figure 5B(b) and in Figure 9". However, when I look at Figure 9, I see absolutely -no- residues labeled on the 2nd loop. The second loop in that figure is actually at the top, and highly solvent exposed according to the figure. Instead, K18 is in the membrane along with the 1st loop. When looking back at Figure 5B(b) at 10ns, it indeed looks like K18 is bringing the 1st loop closely to the CL molecule. However, in the same figure at 20ns, there are -2- residues shown in spherical representations, but only one of them is labeled. The left one is labeled K18, but it looks instead like maybe those would be K44 and K35 instead? It looks like the first loop is now oriented completely away from the bilayer, which would mean that the K18 label is incorrect in Figure 5B(b) at 20ns. Also, did the other simulations show a similar result? Positives: I am fine with the idea that the Autodock data was the primary impetus for the orientations in Figures 9 and 10. Negatives: I am still concerned about the labeling on Fig 5B(b), but I can't verify it myself. It appears that the 1st loop is not in contact with the bilayer. Given the provided orientation at 20ns in Fig. 5B(b), I can't see how K18 can actually be close to the bilayer if Figures 9 and 10 are labeled correctly. More importantly, the rest of the images/data from the remaining simulations need to be reported. There are now 3 more simulations and those data need to be presented to see if they really support the putative orientations. One simulation image just isn't enough in my opinion even if everything is correct in Fig 5B(b). Authors' Response: We have incorrectly stated that K18 is located on the extension of 2nd loop. We should’ve stated that K18 is located on a hoop between loops 1 and 2. The reviewer is also correct that we incorrectly labeled the residues in Fig 5B(b) at 20ns as we believed that CTI makes 90 degrees turn in orientation with a rotation along the long axis from 10ns to 20ns which made us to believe that lysine on the left side at 20ns is K18. Following reviewer’s persistent concern, we carefully inspected amino acid residues in Fig 5B(b) at 20ns and we determined that there are two amino acid residues on the left, K35 and R36, and the residue on the right is K44. We also determined that CTI makes 270 degrees turn in its long axis orientation from 10ns to 20ns which makes CTI to have its 2nd and 3rd loops submerged into membrane, and not solvent exposed as we originally believed. We have made appropriate corrections in labeling the residues in Fig 5B(b) at 20ns and we also corrected orientation of CTI in Fig. 9 and Fig. 10. We also made appropriate changes in the manuscript text which is shaded blue. We thank the reviewer for the sharp eyes, persistence and diligence in reviewing our manuscript which helped to remedy our inaccurate understanding about the orientation of CTI along the membrane surface. Importantly, we have now included a supplementary figure S4 showing additional MD snapshots of CTI/CTII on membrane surface of systems with CL. There is seen to be deeper embedding of CTII in the bilayer, compared to CTI in these 50ns snapshots.
Overall, the new data suggests that CTI and CTII can assume additional orientations with respect to the lipid bilayer and this orientation depends on how many CL residues each protein can interact in the lipid bilayer but we elude to the initial orientations in our paper as one example by which each cardiotoxin can approach the simulated outer mitochondrial membrane and to help build our conceptual mode.
- The manuscript now addresses all of my concerns on my fifth point.
Reviewer 3 Report
Minor revision
Figure 11: It is recommended to add numbers 23 and 35 with felicity, in addition to add the arrows at the positions of K23 and K35.
Author Response
We thank the reviewer for this observation. In response to this concern, we have now made the appropriate revisions to Figure 11 as requested.